# Evaluation of a new rapid diagnostic test, based on the chimeric protein Q5, for the diagnosis of human and canine forms of visceral leishmaniasis

Wagner José Tenório dos Santos[1/+], Natalia Rocha Nadaes[1], Allana Kelly Oliveira Dutra[1], Adalucia da Silva[2], Hemilly Rayanne Ferreira da Silva[2], Artur Leonel de Castro Neto[2], Virgínia Maria Barros de Lorena[2], Valeria Rêgo Alves Pereira[2], Allana Maria de Souza Pereira[2], Maria Edileuza Felinto de Brito[2], Milena de Paiva Cavalcanti[2], Zulma Maria Medeiros[2], Kamily Fagundes Pussi[3], Herintha Coeto Neitzke-Abreu[3], Valeria Marçal Felix de Lima[4], Carlos Henrique Nery Costa[5], Keila Gisele Azevedo F dos Santos[1], Edimilson Domingos da Silva[1], Osvaldo Pompilio de Melo Neto[2/+]

[1]Fundação Oswaldo Cruz-Fiocruz, Instituto de Tecnologia em Imunobiológicos, Bio-Manguinhos, Rio de Janeiro, RJ, Brasil
[2]Fundação Oswaldo Cruz-Fiocruz, Instituto Aggeu Magalhães, Recife, PE, Brasil
[3]Universidade Federal de Grande Dourados, Dourados, MS, Brasil
[4]Universidade Estadual Paulista, Araçatuba, SP, Brasil
[5]Universidade Federal do Piauí, Teresina, PI, Brasil

**BACKGROUND** Visceral leishmaniasis (VL), caused by *Leishmania infantum*, is the most severe form of leishmaniasis, prevalent in many countries, but still with limitations in diagnosis for both human and canine hosts. Serological assays based on recombinant proteins are the most efficient diagnostic alternatives, with the rapid diagnostic test (RDT) being the most cost-effective. The previously described chimeric Q5 is a recombinant protein derived from three native *L. infantum* antigens, which is potentially useful for both human and canine VL diagnosis, through preliminary enzyme-linked immunosorbent assay (ELISA), but which was not evaluated within an RDT setting.

**OBJECTIVES** To evaluate the diagnostic performance of the chimeric recombinant protein Q5 in both ELISA and RDT formats for the detection of human and canine VL, and to compare its performance with RDTs based on Lci2 and Lci13 antigens.

**METHODS** Here, we first expanded the Q5 evaluation through ELISA with a larger set of human and canine VL-positive sera from multiple origins. We confirmed a sensitivity ranging between 80% and 90% for the human VL and greater than 90% with the canine VL sera. A new RDT-Q5 was then set up and tested with multiple batches of human and canine sera.

**FINDINGS** An improved performance was seen for the human VL diagnosis (94% sensitivity), but it was reduced with canine sera (86% sensitivity). Specificity values for both the ELISA-Q5 and RDT-Q5 were generally greater than 95%, with limited (8%) or no false-positive results with human sera from individuals with cutaneous leishmaniasis (CL) and Chagas disease (CD), respectively. The RDT-Q5 performance was compared with RDTs based on two other recombinant proteins, Lci2 and Lci13, tested respectively for the human and canine VL diagnosis.

**MAIN CONCLUSIONS** Despite an equivalent performance for the human VL diagnosis, the RDT-Lci2 led to a much greater incidence of false-positive results with the CL and CD sera. In contrast, no setup for the RDT-Lci13 was effective with the canine sera. Our results confirm the RDT-Q5 as an efficient alternative for the VL diagnosis in the field, particularly for the human form of the disease.

Key words: RDT - ELISA - visceral leishmaniasis - recombinant protein

Visceral leishmaniasis (VL) is the most severe form of leishmaniasis, still prevalent in many developing countries and frequently associated with high morbidity and mortality cases. The protozoan *Leishmania infantum* is the agent responsible for VL in Latin America, where VL is considered a zoonosis, with animal hosts including domestic dogs and, less frequently, other mammals.[1,2] The World Health Organisation (WHO) included VL as a neglected tropical disease, requiring, as targets for 2030, the development of new tools for its prevention, diagnosis, and treatment. Sixty-five of the 70 countries where VL is endemic have been validated for possible VL elimination as a public health problem by 2030. These include Brazil, Ethiopia, India, Kenya, So-

**doi:** 10.1590/0074-02760250126
**Financial support:** FACEPE, CNPq, FIOTEC.
**+ Corresponding authors:** wagner.tenorio@bio.fiocruz.br | ![ORCID] https://orcid.org/0000-0001-7605-4687
osvaldo.pompilio@fiocruz.br | ![ORCID] https://orcid.org/0000-0001-5402-7346

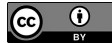

malia, South Sudan, and Sudan, where more than 90% of the VL cases reported worldwide are found.[3] The current context shows the need for developing more sensitive and rapid diagnostic tests for early VL detection in humans as well as in animal reservoirs, particularly dogs, to prevent its spread.

Parasitological diagnosis is considered the gold standard for the identification of hosts infected with *Leishmania*. It requires invasive collection procedures, however, and it is time-consuming, with sensitivity depending on the presence of parasites in the collected samples. Various serological assays have been developed for VL diagnosis, based on the detection in serum or blood samples of antibodies, mainly IgG and IgM, against *Leishmania* proteins. These assays are generally restricted to specific hosts, being dependent on the use of secondary antibodies, such as anti-human or anti-canine IgG, conjugated to an enzyme or fluorescent molecule, for easy detection. Assays available for VL diagnosis include the indirect fluorescent antibody test (IFAT), the enzyme-linked immunosorbent assay (ELISA), the immunoblotting, and the immunochromatographic (IC) or rapid diagnostic test (RDT).[4,5,6,7,8,9]

Serological diagnostic assays based on recombinant proteins, as compared to similar assays using crude or whole *Leishmania* antigens, are easier to produce, cost-effective, and can potentially lead to lower cross-reactions with VL-related diseases.[3,5,10,11] Most used assays for human VL diagnosis are based on the rK39 recombinant antigen, derived from a *L. infantum* kinesin,[12] but although the commercially available RDTs based on rK39 were early on seen to be very efficient in the Indian subcontinent, a lower performance was reported elsewhere.[13,14,15] A chimeric recombinant antigen, rK28, consisting of fragments from three different *Leishmania* antigens, including rK39, has been subsequently produced and found to have a superior diagnostic performance through both ELISA and RDTs with sera from Sudan.[16,17] Another kinesin derivative, more recently developed, found to have a very significant performance for the human VL diagnosis is the recombinant kDDR.[18] Despite the available antigens, however, current strategies for large-scale and cost-efficient VL diagnosis would still benefit significantly from the identification of alternative recombinant antigens with possibly greater efficiency under less favourable conditions, such as in asymptomatic or immunocompromised individuals.

Rapid tests based on rK39 have also been evaluated for the canine VL diagnosis but have been found with variable or suboptimal performances.[6] More efficient results were seen with a synthetic protein derived from the fusion of rK39 with the rK26 antigen, derived from a hydrophilic repetitive protein.[19] This antigen was used to develop a Dual-Path Platform (DPP) test,[20] which is currently recommended for the VL diagnosis in dogs in Brazil. This assay has been assessed through several studies, which confirm its usefulness but highlight the need for further improvements in performance.[21,22,23,24,25] Many other recombinant antigens have also been evaluated for the canine diagnosis, mostly through

ELISA assays but, despite promising results, few have been thoroughly investigated or evaluated as the basis for rapid IC tests.[6,11,18,26,27,28]

Serological assays optimised with the use of antibody-binding proteins, such as protein A (from *Staphylococcus aureus*), can potentially be used for the VL diagnosis from multiple hosts. Protein A exhibits high affinity for several classes and subclasses of antibodies, such as IgG (mostly) and IgM, from several species, including humans, rabbits, and dogs, despite having only a weak interaction with bovine and mouse antibodies.[29] To be effective, however, such assays require recombinant *Leishmania* antigens that are efficiently recognised by antibodies from multiple species. This is a limiting condition, also observed in previous studies by some of us, which identified several new recombinant antigens potentially useful for the serological VL diagnosis. Best antigens for human VL diagnosis, such as Lci2, derived from the same gene encoding rK39, did not perform well with the canine sera. In contrast, those with the best performance for dogs, including Lci3, Lci12, and Lci13, were not efficient with the human sera.[30,31] To produce a single recombinant protein potentially useful for the diagnosis of both human and canine VL, fragments derived from three of the best antigens previously identified (Lci2, Lci3, and Lci12) were joined into new chimeric polypeptides. These were then evaluated through ELISA assays regarding their potential use for VL serodiagnosis with human and canine sera. The best chimeric protein (PQ), named Q5, showed efficient sensitivity results in sera from humans (82%, N = 50) and dogs (100%, N = 39), with no positive reactions with healthy control samples.[32]

In the present study, the recombinant Q5 protein was evaluated using a larger panel of human and canine sera through separate ELISA assays, specific for each host, as well as a protein A-based RDT, applicable to both humans and dogs. The ELISA-Q5 confirmed the efficient performance seen previously with both human and canine VL, while the Q5-based RDT demonstrated equivalent or superior sensitivity and specificity compared to current diagnostic tests for both hosts. Based on these findings, we hypothesise that combining the Q5-based RDT with the ELISA may further enhance diagnostic robustness. The use of RDT-Q5 as a screening test with the rELISA-Q5 having a confirmatory role, for example, may provide a complementary approach that improves accuracy, particularly in diverse clinical settings or in cases with borderline results.

### MATERIAL AND METHODS

*Human sera and ethical considerations (CAEE)* - A total of 102 human sera with a previously positive VL diagnosis were used in the current study Supplementary data (Table). A first set of 41 positive sera was derived from parasitologically confirmed individuals, from the Brazilian State of Piauí (PI) (Protocol CAEE: 0116/2005). These were previously described and used in the first evaluation of the Q5 protein through ELISA.[32] Another set of VL-positive sera included 30 sera from the State of Mato Grosso do Sul (MS), all with a

polymerase chain reaction (PCR) positive result plus parasitological confirmation or at least one other positive result from an independent serological test, such as the ELISA-rK39, RDT-rK39, IFAT or ELISA with whole parasite extract (their use was ethically approved according to the protocol CAEE55556916.2.0000.5160). Similarly, a third sera set, from Pernambuco (PE) state, includes nine with a positive parasitological test plus 22 additional sera tested through PCR or the serological VL tests described above from MS, with a minimum of two positive results (their use approved by the protocol CAEE 0121.0.095.00-08). A total of 64 negative sera from the PE State were selected with negative parasitological tests and negative results in at least two of three serological tests (ELISA-rK39, RDT-rK39, ELISA with soluble parasite protein). Sera from individuals with confirmed cutaneous leishmaniasis (CL) (25 sera) or Chagas disease (CD) (10 sera) were also used. To complete the studies, a last batch of sera was evaluated by the Enrique Dias Foundation (FUNED) from the state of Minas Gerais (MG) (48 positive and 52 negative sera) (Fig. 1A).

*Canine sera and ethical considerations* - A total of 185 VL-positive canine sera Supplementary data (Table), all from parasitologically confirmed dogs, were used (Fig. 1B): 100 from the MS State (Protocol: Nº27/2016); 48 from the State of São Paulo (SP) (Protocol: Nº16/2014); and 37 from the State of Bahia (BA) (Protocol: Nº40/2005), these also used in the preliminary assessment of Q5 through ELISA.[32] In addition, 13 sera were used from PE (Protocol: N° 76/2014) from dogs with the VL diagnosis confirmed through positive results from at least three independent serological assays: DPP, IFAT, and ELISA SLA, based on soluble *Leishmania* antigen. A total of 47 sera were also used, with negative results from both DPP and ELISA SLA: 30 from the PE State, all of which were also negative in the

ELISA-rK39; and 17 from the State of BA. The origins of all the sera used here and in the following topics are represented in (Fig. 1A).

*Expression and purification of recombinant proteins* - Expression of the Q5 protein was carried out as previously described,[32] but with bacterial growth at 37ºC. Protein purification was performed through affinity chromatography with the AKTA Pure (Cytiva) set up, with buffer A (20 mM Tris-HCL, 8 M urea, 20 mM imidazole and 250 mM NaCl pH 8.0) and buffer B (20 mM Tris, 8 M urea, 500 mM imidazole, and 250 mM NaCl pH 8.0), used respectively for washes and elution. The recombinant proteins Lci2 and Lci13 were expressed in *Escherichia coli* (BL21 (DE3)) and purified as described in previous studies,[30,31] in the presence of 8 M urea.

*ELISA assays using recombinant proteins (rELISA)* - The methodology used for the ELISA tests was already described,[32] where a dilution of 1:2500 was used for the human sera, and 1:900 for the dog sera, and the tests were performed with the Q5 protein purified in 8 M urea (6 µg/mL). For the ELISA with the human and canine sera, the peroxidase conjugated goat anti-human IgG (Sigma-Aldrich diluted 1:10.000) or anti-dog IgG (Sigma-Aldrich diluted 1:1.200) was used, respectively.

*RDT based on the recombinant* - For the RDT prototypes, the purified recombinant proteins (Q5, Lci2, or Lci13) were impregnated on nitrocellulose membranes, also impregnated in parallel with the protein A control, following the same parameters of a previous study.[33] All RDTs tested here were based on the following set-up sequence: sample pad, conjugate pad, nitrocellulose membrane impregnated with the antigen (for the test line) and the control, followed by the sink pad (to absorb residues from the run). These were cut into strips, 5.2 mm wide, and placed in plastic cassettes. After various analyses, the final test conditions for the

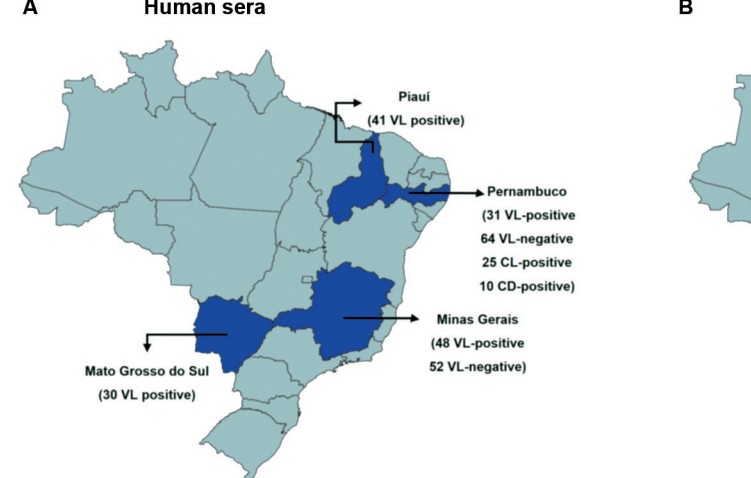
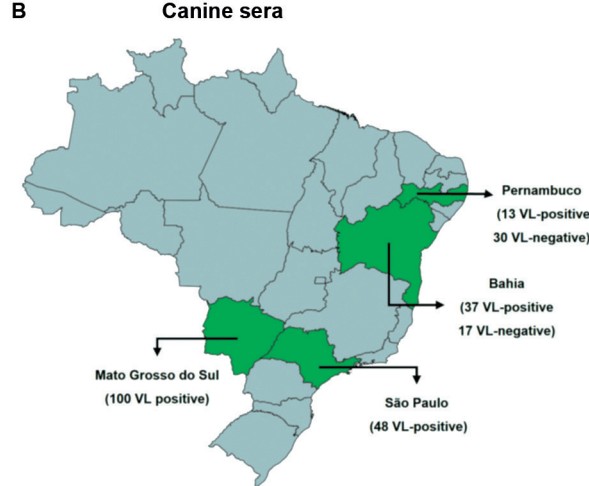

Fig. 1: description of human and canine serum panels employed in this study: geographic distribution in Brazil, highlighting the states where human (blue) and canine (green) serum samples were collected. These samples were used throughout the study, except for the visceral leishmaniasis (VL)-positive and VL-negative human sera from Minas Gerais (MG), which were exclusively tested with the rapid diagnostic test (RDT)-Q5 by Enrique Dias Foundation (FUNED)-Laboratório Central (LACEN).

optimised RDT-Q5 required the impregnation with 0.1 microliter of protein per millimetre of membrane. These tests used protein A for serum detection, and the same tests were used for the sera from both humans and dogs. The test procedures were standardised so that five (serum) or ten (blood) microliters of the sample could be added to each well, followed by the addition of three drops of running buffer to allow the sample fluid to flow through capillary action along the length of the nitrocellulose membrane. The results were determined visually after 20 min of incubation at room temperature (average 22ºC). Positive results were indicated by the appearance of bands coloured pink or purple on the test and control lines, while only the control line appeared for negative results. If the control line did not appear, the test was considered invalid. Tests were then set up for different batches of recombinant proteins. The RDT cassettes were stored at room temperature in silica-protected containers until use.

*Statistical analyses* - Sensitivity, specificity, and confidence interval parameters were estimated with the software MedCalc (version 12.3) (MedCalc Software, Ostend, Belgium). The GraphPad Prism program was used to generate the dot plot, the receiver-operating characteristics (ROC) curve, and the bar chart (GraphPad Prism version 6.00 for Windows, GraphPad Software, La Jolla, California, USA). The cut-off was determined by adding twice the standard deviation of the negative sera optical density to the mean optical density of the negative samples, corresponding to a 95% confidence interval.

## RESULTS

*ELISA-Q5 for the diagnosis of human VL* - Thirty-one sera from PE were first tested, with six producing false negative results and defining a sensitivity of 81%. Another 30 sera from MS were also tested, with three false negative results and 90% sensitivity (cutoff 0.089). To determine specificity, 64 VL-negative sera were also tested, with only one producing a false positive result (Fig. 2A-B). ROC curves were next generated comparing the performance of the ELISA-Q5 with the sera from the two Brazilian states and confirming its greater performance with the sera from MS (Fig. 2C). Combining the data for the entire set of tested sera revealed an overall sensitivity of 85%, with 98% specificity. Next, we tested twenty-five sera from patients with CL, with only two sera testing positive (8%), followed by testing of ten sera from individuals with CD, with no positive results. Overall, these results confirm a very good performance of the ELISA-Q5 for the human VL diagnosis.

*ELISA-Q5 for the canine VL diagnosis* - In this study, we reassessed the ELISA-Q5 using a significantly larger sample of VL-positive canine sera (summarised in Fig. 1B). We first tested two sets of parasitologically confirmed sera from two different Brazilian states: 100 sera from MS and 48 sera from SP (all ELISA-Q5 results with the canine sera shown in Fig. 3A-B). For the MS sera, only one negative result was observed, yielding a sensitivity of 99% for the assay. In contrast, the SP sera showed a sensitivity of 92%, with four false-negative results. A third set of VL-positive sera, from PE, included

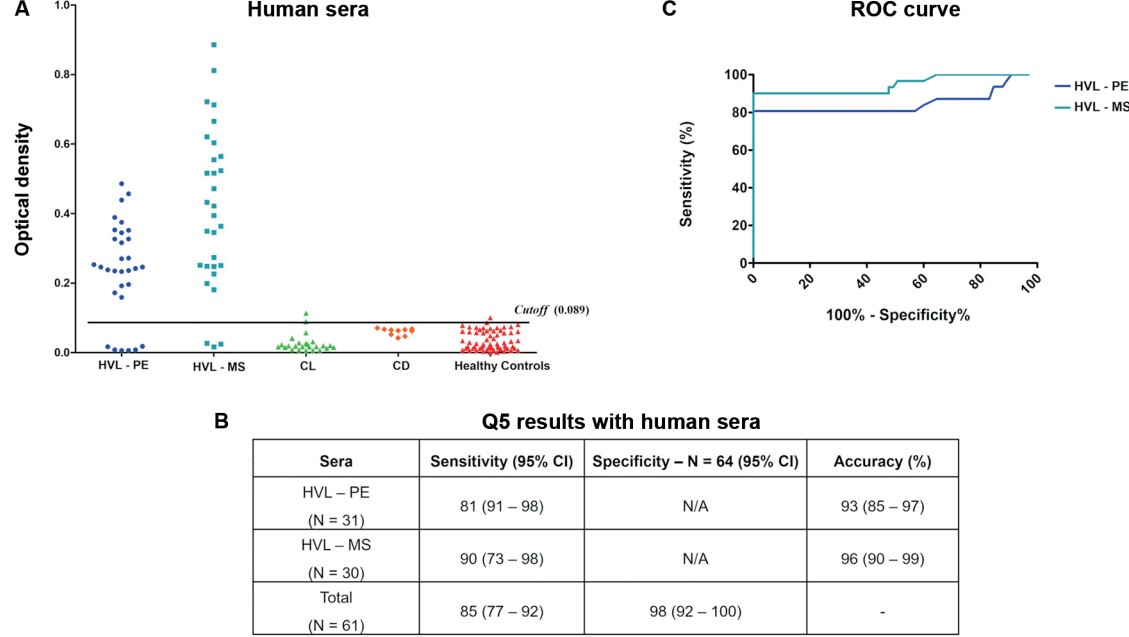

| Sera | Sensitivity (95% CI) | Specificity – N = 64 (95% CI) | Accuracy (%) |
|---|---|---|---|
| HVL – PE (N = 31) | 81 (91 − 98) | N/A | 93 (85 − 97) |
| HVL – MS (N = 30) | 90 (73 − 98) | N/A | 96 (90 − 99) |
| Total (N = 61) | 85 (77 − 92) | 98 (92 − 100) | - |

Fig. 2: evaluation of the enzyme-linked immunosorbent assay (ELISA)-Q5 protein with human sera. (A) ELISA-Q5 results after testing with the human visceral leishmaniasis (HVL) sera from Pernambuco (PE) and Mato Grosso do Sul (MS). (B) Summary of the ELISA-Q5 results with the defined values for the sensitivity, specificity, and accuracy parameters is shown, with the 95% confidence intervals (CI) indicated in parentheses. (C) Receiver-operating characteristics (ROC) curve analysis based on the ELISA results shown. Each dot in the ELISA assay represents the average of a technical triplicate.

13 sera diagnosed as positive by both EIE-LVC and DPP tests. Among these, two negative results were recorded with ELISA-Q5, resulting in a sensitivity of 85%. To determine specificity, we also tested 47 sera with a previously defined negative diagnosis, of which only one was a false positive (cutoff 0.286). Separate ROC curves were also generated for all three sera sets, showing the equivalent performance of the ELISA-Q5 with all (Fig. 3C). Overall, based on the 161 positive sera tested, the ELISA-Q5 demonstrated 96% sensitivity and 98% specificity for diagnosing VL in dogs, consistent with the excellent performance originally defined for this assay.

*RDT based on the recombinant Q5* - Considering the significant ELISA-Q5 performance, we next aimed to evaluate its performance as part of an immunochromatographic RDT test based on a lateral flow platform. The RDT used here, represented in Fig. 4A, allows loading of either sera or blood samples into wells (Fig. 4B), with a positive result indicated by the appearance of bands on the test and control lines (Fig. 4C). To compare the Q5 in the RDT with another recombinant antigen with similar performance in the ELISA assay for the human VL diagnosis, we also set up an experimental RDT prototype based on the Lci2 antigen. Likewise, a third RDT was set up with the recombinant Lci13, aiming for a comparative analysis with the canine sera only.

*Evaluation of the RDT-Q5 for the human VL diagnosis* - For a first assessment of the RDT-Q5 applied for the human VL diagnosis, a first batch was produced (Q5/2017) and tested, with representative positive and negative results shown in Fig. 5A. The RDT-Q5 was first evaluated with a limited panel of 41 VL-positive human sera from Piaui, with only two of those sera producing a false negative result. Ten negative sera, from Pernambuco, were also tested, with two producing a false positive result, with the preliminary sensitivity and specificity values defined at 95% and 80%, respectively (Fig. 5B). A second batch of the RDT-Q5 test was then produced at similar conditions (Q5/2019) and assessed with a total of 102 VL-positive sera from three Brazilian states (PI, MS and PE), with six producing false negative results (94% sensitivity). A total of 64 sera from negative controls were also tested, but only two sera showed a positive result (97% specificity). This test was also assessed with sera from 25 individuals afflicted with CL and ten with CD (Fig. 5A). Only two of the CL-positive sera produced positive results, indicating low cross-reactivity with the cutaneous form of the disease and no cross-reaction with the CD sera, even with the very effective performance with the human VL sera.

Aiming to improve further the performance of the tests, a third batch was produced with new setup conditions (Q5/2021), which included the recombinant Q5 purified using 2 M urea (all other protein purifications used for the assays described here used 8 M urea). This new batch was evaluated with a selection of 45 VL-positive and 45 VL-negative sera from those tested with the Q5/2019 batch, with only one serum producing a false negative result (96% sensitivity/100% specificity over-

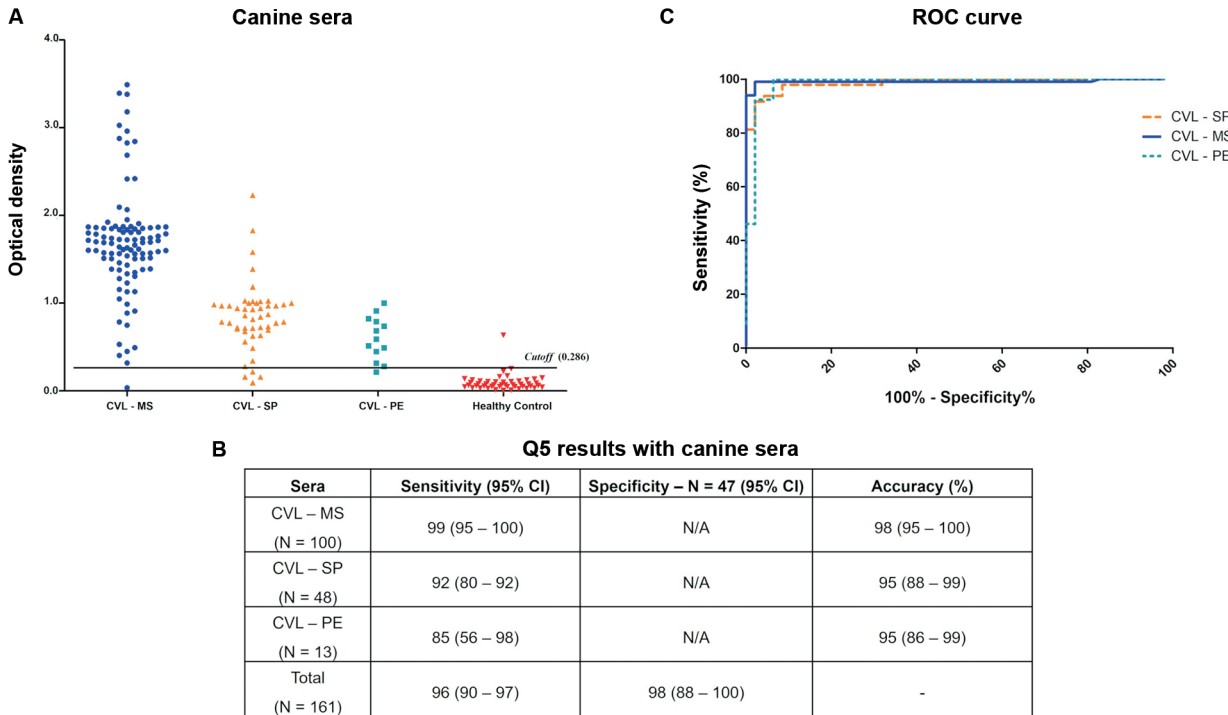

Fig. 3: evaluation of the enzyme-linked immunosorbent assay (ELISA)-Q5 protein with canine sera. (A) ELISA-Q5 results after testing with the canine visceral leishmaniasis (CVL) sera from Pernambuco (PE), Mato Grosso do Sul (MS), and São Paulo (SP). (B) Summary of the ELISA-Q5 results with the defined values for sensitivity, specificity, and accuracy parameters is shown, with the 95% confidence intervals (CI) indicated in parentheses. (C) Receiver-operating characteristics (ROC) curve analysis based on the ELISA results shown. Each dot in the ELISA assay represents the average of a technical triplicate.

all). The Q5/2021 batch was also sent to an independent laboratory [FUNED-Laboratório Central (LACEN), from the State of MG], where values of 98% sensitivity (N = 48) and 90% specificity (N = 52) were observed. These results confirm the reproducibility of the RDT tests based on the chimeric Q5 but do not show any substantial improvement with the lower urea concentration.

We next opted to compare the performance of the RDT-Q5 with an equivalent test based on the Lci2 antigen. The RDT-Lci2 was then evaluated with a limited number of sera: 45 sera from VL-positive individuals, 40 sera from negative controls, 10 sera for CL, and 10 sera for CD. The results showed a sensitivity of 96% and a specificity of 93% (Fig. 5B-C), but cross-reactions were seen not only with sera from individuals with CL, with six positive results, but also with CD sera, with four of those producing a positive result. Despite a similar efficiency for the VL-diagnosis with the human sera, when compared to the RDT-Q5, the RDT-Lci2 is therefore less able to avoid cross-reactions or false positive results with the closely related diseases.

*Evaluation of the RDT-Q5 for the VL diagnosis with canine sera* - All the RDT tests used protein-A for the detection of VL-positive antibodies and could potentially be used not only with human sera but also with sera from different animal species. For a preliminary evaluation of the RDT-Q5 for the canine VL diagnosis, the first batch produced (Q5/2017) was also assessed with a limited panel of 38 sera from confirmed VL-positive dogs, as well as from 12 negative control animals, from Bahia. Representative positive and negative results are shown in Fig. 6A, with only one false-negative and no false-positive results observed, leading to 97% sensitivity and 100% specificity values (Fig. 6B). Another assessment was performed with the second RDT-Q5 batch produced (Q5/2019) and a larger set of 198 VL-positive and 47 VL-negative sera from four Brazilian states, including those sera already assessed with the Q5/2017 batch. The results, however,

indicated a poorer performance for the test, with 86% sensitivity and 96% specificity. With the changes made to the third RDT-Q5 batch described above (Q5/2021), yet another evaluation was carried out with a subset of 82 VL-positive and 17 VL-negative canine sera, with observed sensitivity and specificity values of 89% and 94%, respectively (also shown in Fig. 6B). These results are consistent with the less efficient performance of the RDT-Q5 with the canine VL sera, compared to its performance for the human VL diagnosis.

Attempts were also made to properly evaluate the RDT-Lci13 for a comparative assessment regarding the canine VL diagnosis. However, despite multiple optimisation steps and modifications regarding membrane types, protein concentrations, and buffers, none of the setups tested were able to bring any efficiency for this test in identifying VL-positive canine sera (Fig. 6C).

*Comparative analysis of the diagnostic performance of Q5 using ELISA and RDT* - Overall, the results shown so far indicate a better performance for the ELISA-Q5 with the sera from VL-positive dogs, while the RDT-Q5 performed better with the human sera, with specificity values for both tests generally found to be greater than 95%. To fully understand this discrepancy, we opted to directly compare the results from the ELISA-Q5 with the second RDT-Q5 batch (Q5/2019), both tested with larger assemblages of sera from multiple Brazilian states for both human and canine sera, as shown in Table. When the human sera are considered, both tests were assessed with the same set of 102 VL-positive sera, including 41 sera from PI, whose results for the ELISA-Q5 were reported in our previous publication.[32]

All six sera producing negative results with the RDT-Q5 were also found to be negative with the ELISA-Q5, but the latter test also produced negative results for nine other sera, confirming the reduced performance seen for the ELISA assay as compared to the immunochromatographic test, but with an overall agreement of 91% (93/102) between the two tests. For the negative

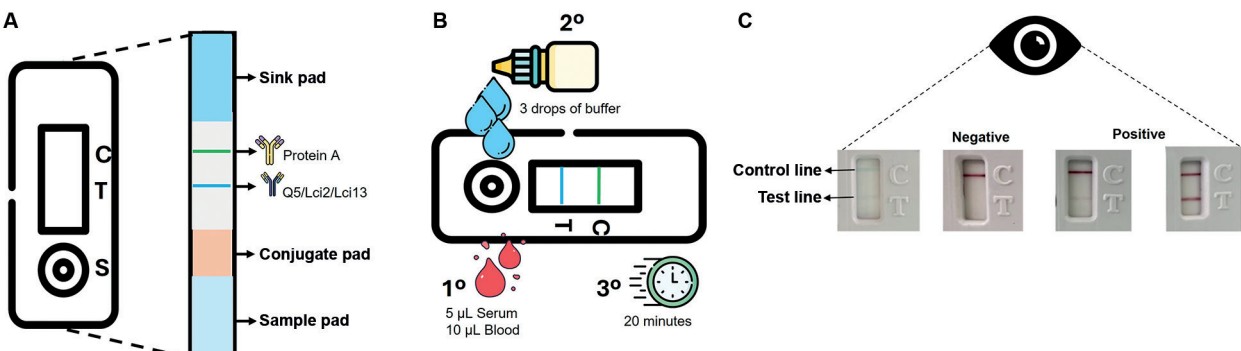

Fig. 4: description of the cassettes used for the rapid diagnostic tests (RDTs) evaluated in this study. (A) The assembled strips found within the cassettes included: a card to hold the buffer and red blood cells (named "Sample pad"); the membrane loaded with colloidal gold ("Conjugate pad"); a nitrocellulose membrane where the different antigens assessed (Q5 or Lci2 or Lci13) were loaded (blue line), as well as the protein A (control, green line); and the waste membrane ("Sink pad"). (B) Five microlitres of serum or ten microlitres of human or canine blood were used for each test, carried out with three subsequent drops of running buffer followed by up to 20 min incubation for the results to appear. (C) The test result may indicate reactivity on the "Test line" (negative or positive, strong or weak) or an invalid test (no signal on the Control line). (C) Control line. T: test line; S: sample well.

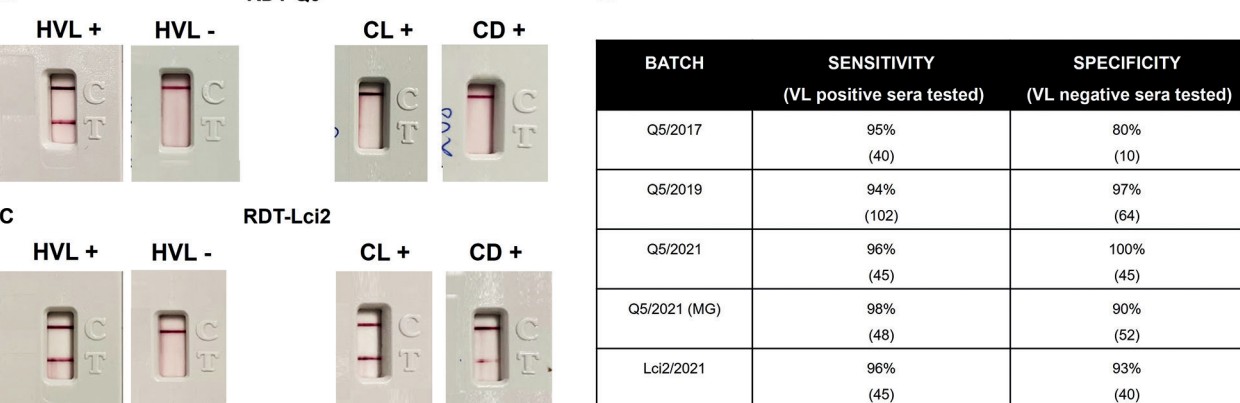

Fig. 5: reactivity of the new rapid diagnostic tests (RDTs) described here with human visceral leishmaniasis (HVL) sera. (A) Representative results showing the reactivity of the RDT-Q5 with positive and negative HVL sera, as well as with sera from individuals with cutaneous leishmaniasis (CL) and Chagas disease (CD). The test and control lines are seen in pink/purple. (B) Summary of the results from testing the various batches of RDTs assessed here. (C) Representative results showing the reactivity of the RDT-Lci2 with VL-positive, VL-negative, CL-positive, and CD-positive sera.

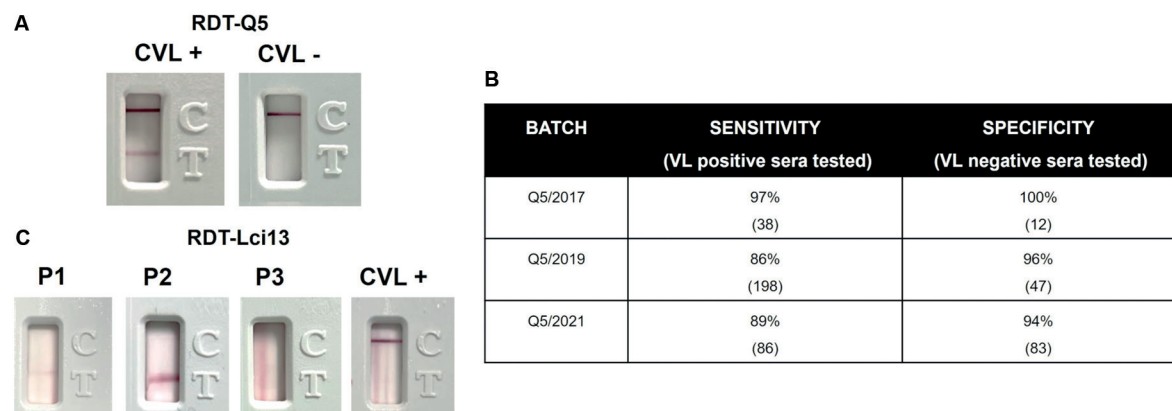

Fig. 6: reactivity of the different batches of the rapid diagnostic test (RDT)-Q5 with canine visceral leishmaniasis (CVL) sera. (A) Representative results showing the reactivity of the RDT-Q5 test with positive and negative CVL sera. (B) Summary of the results from the testing of the various batches of the RDT-Q5 assessed here. (C) Representative results showing the reactivity of various setups for the RDT-Lci13. Several of those tested various buffers and protein concentrations, but the test line, or smears, appeared in the absence of the canine VL-positive sera (P1, P2, P3). With other setups, the test line did not appear even after testing with multiple sera from VL-positive dogs (CVL+).

control sera, three of those produced contrasting results, from a total of 64, with a similar agreement overall (95%). For the VL-positive canine sera, the two tests were also assessed with the same set of 198 VL-positive canine sera, including sera from BA, which were included in the previous analysis of the ELISA-Q5. Eleven were found to be negative for both tests, with only one test found to be negative with the ELISA-Q5, but positive for the RDT-Q5, while 17 sera, which were found negative for the rapid test, produced positive results using the ELISA-Q5 (~91% agreement overall). For the VL-negative canine sera, from a total of 47, an agreement of ~98% was observed between the two tests, with only one serum producing a false positive result for the RDT-Q5 only, and another producing a false positive result for both the ELISA-Q5 and RDT-Q5. This analysis confirms the substantial agreement between the two tests based on the Q5 protein.

## DISCUSSION

The current assessment of the chimeric Q5 expands on its previous evaluation as the basis of a novel ELISA assay with possible use for the VL diagnosis.[32] The same assay was used more recently to evaluate sera from dogs diagnosed with VL based on commercial serological tests currently recommended by public health authorities, namely the EIE-LVC and DPP tests, but the results regarding the efficiency of the ELISA-Q5 were mainly inconclusive.[34] Here, assessing sera from multiple localities and in greater numbers, we were able to confirm the versatility of using this PQ for the VL diagnosis in both humans and dogs, with a noticeably better performance for the canine sera. This versatility is supported by the results using the RDT-Q5, first described in the current study, which was based on a setup with another antigen previously tested for VL diagnosis in dogs.[35] Our results

TABLE

Agreement between the performance of enzyme-linked immunosorbent assay (ELISA)-Q5 and rapid diagnostic test (RDT)-Q5 using visceral leishmaniasis (VL)-positive and human VL (HVL)-negative and canine VL (CVL) sera

|  |  | ELISA-Q5 + | ELISA-Q5 - | Total | Agreement |
|---|---|---|---|---|---|
| HVL positive | RDT-Q5 + | 87 | 9 | 96 | 91,17% (93/102) |
|  | RDT-Q5 - | 0 | 6 | 6 |  |
|  | Total | 87 | 15 | 102 |  |
| HVL negative | RDT-Q5 + | 0 | 2 | 2 | 95,31% (61/64) |
|  | RDT-Q5 - | 1 | 61 | 62 |  |
|  | Total | 62 | 2 | 64 |  |
| CVL positive | RDT-Q5 + | 169 | 1 | 170 | 90,91% (180/198) |
|  | RDT-Q5 - | 17 | 11 | 28 |  |
|  | Total | 186 | 12 | 198 |  |
| CVL negative | RDT-Q5 + | 1 | 1 | 2 | 97,87% (46/47) |
|  | RDT-Q5 - | 0 | 45 | 45 |  |
|  | Total | 1 | 46 | 47 |  |

Number of sera with positive (+) or negative (-) results with either the ELISA-Q5 or the RDT-Q5, with the agreement values (total number of sera with the same results in both tests/total number of sera assessed).

show a very efficient and consistent performance for the RDT-Q5 diagnosis regarding the human form of the disease, while showing its potential use also with the canine VL, but which would still need some improvement.

The need for new diagnostic tools that are rapid, cost-effective, and efficient, for the monitoring and treatment of human and canine VL is highlighted by the current dependency of public health authorities on tests still based on crude native proteins extracted from cultured parasites. Current assays are also lacking in efficiency for the diagnosis of immunocompromised individuals, such as those co-infected with human immunodeficiency virus (HIV), or impacted by age,[5,36,37,38] and asymptomatic cases, mainly in dogs.[6,9] Indeed, several recent publications still investigate or describe new antigens aiming at their potential use for serological VL diagnosis,[26,27,28,39-43] confirming the need for further work aiming to improve current diagnostic methods.

Early studies considered using chimeric antigens to maximise their potential use for VL diagnosis using ELISA assays. An example is the PQ, derived from ribosomal proteins and histones, but with a lower than 80% sensitivity for the VL diagnosis in dogs.[44] PQs based on the K9/K26/K39 antigens were later evaluated, and a greater performance (~95% sensitivity) was seen for the canine VL,[19] with a related protein currently being the basis for the DPP test.[20] A similar performance with canine VL was seen for the synthetic, multiepitope, PQ10[26,45] and, as seen here for Q5, the recombinant PQ10 had a lower, but still relevant, efficiency with human VL sera (>80% sensitivity).[42] An attempt for a better PQ to be used for the human VL diagnosis was the synthetic glucose-regulated protein 78 (GRP78), ubiquitin-conjugating enzyme E2, calreticulin, mitochondrial heat shock 70-related protein 1 (mtHSP70) (GRP-UBI-HSP), another multiepitope protein, but a limited sensitivity (~70%) was observed for the recombinant protein.[41] The chimeric Q5 seems to be at least as efficient as the best PQ previously assessed for the diagnosis of both human and canine VL, reinforcing it as a viable alternative for the VL diagnosis, with potential use for both forms of the disease with for the diagnosis of human or canine VL. It contrasts with the performance of the individual proteins from which it was derived, either effective for the diagnosis of human or canine VL but not for both, and validates the approach used to improve its efficiency.[32]

The performance of the immunochromatographic RDT-Q5 with the human VL sera qualifies it as a cost-effective alternative for the diagnosis of human infection. Nevertheless, the comparative analysis between the ELISA and RDT assays indicates differences in sensitivity between the two tests regarding the canine and human sera, which are not directly related to the antigen used. Differences in sensitivity between ELISA and rapid test assays have also been reported for the synthetic rKD-DR-plus antigen, derived from rK39, when both were evaluated for the human VL diagnosis, but with the ELISA having a better performance.[43] These differences might be attributed to the use of a microtiter plate compared to a nitrocellulose membrane, secondary antibodies, or even blocking reagents. Alternatively, improvements on the recombinant antigen, possibly through the enhancement of its antigenic diversity, might further increase the efficiency of either or both tests and facilitate their use for both human and canine VL.

VL and HIV-acquired immunodeficiency syndrome (AIDS) coinfection is considered a life-threatening pathology when undiagnosed and untreated, due to the im-

munosuppression caused by both diseases.[37] Commercially available tests currently used in Brazil for human VL diagnosis are based on rK39, which is not effective for immunocompromised individuals.[46] The recombinant Lci2, one of the three proteins whose fragments were used to produce the chimeric Q5, is more effective for the VL diagnosis of VL/HIV co-infections,[47] so a possible use of the RDT-Q5 in these situations can also be considered. Likewise, asymptomatic dogs are fully competent for VL transmission[48] despite having lower antibody titres, leading to an impaired diagnosis by serological methods.[6,9,20,49] The chimeric PQ10 antigen and the related PQ20 were specifically investigated regarding their potential for the diagnosis of naturally and experimentally infected dogs. Both were more efficient in identifying asymptomatic dogs infected with *L. infantum* than a serological assay with crude *Leishmania* antigens, with the results from the immunoassay relating to parasite load.[27,42]

The DPP® technology, used in the currently recommended diagnostic test for canine VL, is characterised by having different openings for loading of the sample (analyte) to be tested and for the buffer solution required for the diagnostic reaction to proceed. The analyte of interest migrates along the first membrane, binding to the capture agents immobilised in the test region. The buffer then allows the labelled ligands (probe) to migrate along a second membrane and bind to the analyte captured in the test region.[50] In contrast, the Lateral Flow procedure used here for the RDT-Q5 consists of a single multimembrane strip arranged sequentially under an adhesive card. The strip is placed inside a plastic cassette where the reaction occurs. This device has a single opening for dispensing the sample/buffer, and its simpler design should be more cost-effective without loss of reliability.[33] The performance of the RDT-Q5 with asymptomatic animals still needs to be evaluated, but based on the results seen so far, it might be considered a relevant alternative for their diagnosis in its current format or after further improvements.

The results shown here then confirm the potential for the chimeric Q5 as a promising new tool for the VL diagnosis of both human and canine forms of the disease. Its performance within the new RDT test, which is most likely to be used in general, validates its use as it is for the human VL diagnosis. Still, the recombinant Q5 can also be evaluated as part of a multiplex test (with several recombinant antigens). Further performance improvement can also be considered, especially for its use in canine VL diagnosis and for other animal species. As previously shown during the Q5 design,[32] this may be achievable with reasonable ease and speed with the inclusion of new antigenic epitopes within its sequence, leading to an RDT equally effective for human and animal diagnosis to be used cost-effectively in the field as well as in laboratory settings. Subsequent studies will require as much as possible the use of sera from individuals with the VL diagnosis confirmed parasitologically, or at least through PCR, to avoid bias introduced by previous testing with other serological methods, and which might partially be a reason for the reduced sensitivity observed here with the ELISA-Q5 for both human and canine samples from the State of PE. A larger number of negative human and canine samples also need to be included, as well as an evaluation of sera from individuals with other associated diseases, to assess cross-reactions. For the evaluation with the canine sera, these shall include sera from animals with confirmed diagnosis for diseases caused by other trypanosomatids, as well as *Ehrlichia canis*, *Babesia canis*, and others. Considering the human sera, a greater number of samples from individuals with CD and CL should be evaluated, as well, sera from individuals afflicted with other diseases. Further validation of the test will require its assessment with whole blood samples under simulated or real field conditions, in order to confirm the test's applicability and its robustness in real-world settings. If the RDT-Q5 performance is confirmed effective for both human and canine VL, the standardised conditions allowing the same rapid test to be used for diagnosing both humans and dogs in field settings should greatly facilitate the streamlined production of a single test for both hosts and improve its cost-effectiveness and widespread use.

## ACKNOWLEDGEMENTS

To the Vice Directorate of Reagents for Diagnostics from Bio-Manguinhos (Fiocruz), especially Dr Antonio Gomes Pinto Ferreira, for their support.

## AUTHORS' CONTRIBUTION

WJTS - conceptualisation, investigation, data curation, writing, review & editing, visualisation (figure creation); NRN - conceptualisation, investigation, data curation; AKOD - investigation and data curation; AS, HRFS, VMBL, VRAP and AMSP - data curation, writing and review & editing; ALCN - investigation, writing and review & editing; MEFB, MPC and ZMM - review & editing; KFP, HCNA, VMFL, CHNC and KGAFS - data curation, writing and review; EDS and OPMN - conceptualisation, project administration & supervision, investigation, data curation, writing and review & editing. All authors hereby declare that they have no conflicts of interest, whether financial or personal, that could have influenced the work reported in this review paper.

## DATA AVAILABILITY

The datasets generated and analysed during the current study are included in this published article and its supplementary material. Additional raw data supporting the findings of this study are available from the corresponding authors upon reasonable request. All sensitivity, specificity, and agreement calculations were based on the data provided in the Supplementary data (Table).

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

# OPEN PEER REVIEW

Memórias do IOC thanks the anonymous reviewers for their contribution to the peer review of this work.

**FIRST REVIEW ROUND**

**REVIEWER #1**

General Comments

I would like to thank you for the opportunity to review this manuscript. The approach, focused on achieving accurate diagnosis, is highly relevant for the control of human and canine visceral leishmaniasis, particularly within the context of public health.

The manuscript presents solid scientific merit, with well-defined results and a clear contribution to improving the diagnosis of leishmaniasis. However, the text would benefit from improved fluency by restructuring long sentences, incorporating clear connectors, and enhancing the logical flow of ideas. It is also recommended that the discussion be expanded to better address the study's limitations, potential improvements to the test, and the practical implications for its field application.

Additionally, it is important to include confidence intervals for the reported sensitivity and specificity values, as these are essential for assessing the robustness and reliability of the results.

In the introduction, it is important to highlight the use of Protein A as a universal conjugate, emphasizing its applicability and performance with both human and canine IgG . Include an explanation of the differences between Lateral Flow (LF) and Dual Path Platform (DPP). It is also essential to underscore a key advantage: the standardized conditions allow the same rapid test to be used for diagnosing both humans and dogs in field settings. This approach streamlines production and improves cost-effectiveness.

Additionally, the complementary role of ELISA and the rapid test (RDT) should be reinforced, as their combined use enhances the accuracy and reliability of Leishmania infantum infection detection. It should be clearly stated which platform is intended for screening and which one is proposed as a confirmatory test, considering their suitability for large-scale surveillance in transmission areas.

Include a table summarizing all the production conditions of the evaluated batches, along with the corresponding serum sample testing results.

Title: Evaluation of a new rapid diagnostic test based on the chimeric protein Q5 for the diagnosis of human and canine visceral leishmaniasis

Keywords: RDT; ELISA; Visceral Leishmaniasis; recombinant protein.

Introduction (Lines 139-144)

In the present study, the recombinant Q5 protein was evaluated using a larger panel of human and canine sera through both ELISA and rapid diagnostic test (RDT). The Q5-based RDT demonstrated equivalent or superior sensitivity and specificity compared to current diagnostic tests for both human and canine visceral leishmaniasis (VL), reinforcing its potential as a reliable tool for VL diagnosis. Based on these findings, we hypothesize that combining the Q5-based RDT with the ELISA may further enhance diagnostic robustness, offering a complementary approach that improves accuracy, particularly in diverse clinical settings or in cases with borderline results.

Fig. 1. Description of human and canine serum panels employed in this study: Geographic distribution in Brazil highlighting the states where human (blue) and canine (green) serum samples were collected. These samples were used throughout the study, except for the VL-positive and VL-negative human sera from Minas Gerais, which were exclusively tested with the RDT-Q5 by FUNED-LACEN.

Fig. 2. – provide an Y-axis label

Results Section:

Please verify whether some content overlaps with the Materials and Methods or Discussion sections, as it appears there may be some mixing. Previously published results should be included only in the general discussion, not in the results section.

I recommend adding an introductory paragraph summarizing the main findings of this study, followed by a clear presentation of the results organized by specific topics. Including a summary table with consolidated data, absolute numbers (XX/XX) and corresponding percentages, is also advisable.

Clarify the evaluation of the test with whole blood samples under real field conditions.

Discussion Section:

The discussion is generally well presented. However, it would benefit from an introductory paragraph summarizing the study's objectives and relevance before discussing the findings in detail.

It is recommended to include a specific evaluation of the test using whole blood under field conditions. This assessment is essential to demonstrate the test's applicability and robustness in real-world settings,

beyond laboratory conditions. Providing this information will strengthen the evidence for the test's operational performance and its suitability for point-of-care use in both human and canine populations.

Further comments are provided directly in the manuscript file.

Sincerely,

**REVIEWER #2**

Evaluation of a new rapid diagnostic test based on the chimeric protein Q5 and applicable for the diagnosis of human and canine forms of visceral leishmaniasis

The manuscript by dos Santos et al. presents a reassessment of the Q5 chimeric recombinant protein using the ELISA platform with a larger set of human and canine VL-positive sera. It also describes the development of a new RDT-Q5, tested with human and canine sera from various states in Brazil. The study is scientifically sound, and the results indicate that Q5 shows promise for the serological diagnosis of VL—through ELISA assays for dogs and RDTs for humans. The article is well-written, and the results adequately support the conclusions. However, the authors must address sime issues.

a) Adequacy of the abstract

The abstract is well-written and provides a clear summary of the study.

b) Originality and importance of the contribution to the field

The study is both relevant and innovative, as it aims to demonstrate the utility of the Q5 chimeric protein on an immunochromatographic platform for the diagnosis of human and canine leishmaniasis.

c) Introduction

On page 6, the paragraph between lines 84–96 begins with the statement: "Clinical canine VL diagnosis can be impaired due to its presentation overlapping with other infections or hematological malignancies." However, the subsequent information applies to both canine and human VL, and most of the cited references pertain to human VL. Furthermore, in human VL, the clinical features are also not pathognomonic, as symptoms overlap with those of other diseases. The authors should revise this paragraph to improve clarity and ensure consistency.

d) Methodology

The methodology is well described. Specific comments:

The authors did not describe the samples from Minas Gerais (MG) in the Materials and Methods section; instead, this information appears in Figure 1. The authors should include a description of these samples in the Materials and Methods section.

Please place the number of the figure at the end of the sentence (line 178).

The authors state in the Statistical Analysis section that Cohen's Kappa (κ) analysis was performed; however, the results of this analysis are not included in the manuscript. This should be addressed.

e) Results

The results are well presented. However, the following points could help improve the presentation:

It is recommended that the authors specify Figure 1A in line 246.

On page 18, lines 353–355: "It was sent to an independent laboratory (FUNED-LACEN, from the state of Minas Gerais Q5/2021 (MG) where values of 98% sensitivity (N=48) and 90% specificity (N=52) were observed." — please close parentheses appropriately.

The authors should review the results for batches Q5/2017 and Q5/2021 in Figure 6B (page 20) and revise the corresponding text in the results section (lines 383–394) for consistency.

The samples from Pernambuco, both human and canine, showed lower sensitivity compared to the other groups. Could the authors comment on whether this might be related to the diagnostic tests used? For dogs, serology was used; for humans, PCR and serology were applied in 22 of the 31 samples.

f) Discussion

In the first paragraph, the authors could enhance the discussion by addressing not only immunosuppression but also age as a key factor associated with the lower accuracy of serological tests for human visceral leishmaniasis (Freire et al., 2019; Cota et al., 2012).

The smaller number of negative controls in both human and canine samples, compared to the positive ones, should be acknowledged as a limitation of the study in the discussion.

It is recommended to include in the discussion the limitation of not incorporating samples from dogs with other diseases—such as trypanosomatids, Ehrlichia canis, Babesia canis, and other Leishmania species—among the truly negative control group.

The discussion should also address the limitation of using a small number of control samples with potential cross-reactivity (e.g., Chagas disease and cutaneous leishmaniasis) and the lack of samples from humans with other diseases, such as paracoccidioidomycosis, malaria, and active pulmonary tuberculosis.

Additionally, the discussion should mention the limitation of using serology rather than parasitological tests as the reference standard to classify the 13 dogs from Pernambuco as positive, and how this could introduce classification bias.

g) References

Current, extensive and strategic references to the theme have been included.

h) Figures and tables

I suggest that the authors include a table with the demographic data of the human patients used in the study.

Does Figure 1A refer to the "blue map"? The authors should ensure consistent labeling between the figure and the text (line 165).

In Figure 2A, the Y-axis should be labeled as "Optical Density."

In Figure 2C, the Y-axis should be labeled as "Sensitivity (%)". Additionally, please verify that the identification of the curves (HVL-MS and HVL-PE) matches those in Figures 2A and 2B.

In Figure 2B, the summary of the ELISA-Q5 results should be presented similarly to how it is shown in Figure 3B. In both figures, the total number of positive samples should be included in the last line. Furthermore, the number of negative samples should be specified, as all figures and tables should be self-explanatory, without requiring the reader to refer back to the text.Conceptually, positive and negative predictive values (PPV and NPV) depend on the prevalence of the disease in the population being tested. When samples are selected based on prior diagnosis, PPV and NPV will vary according to the proportion of positive and negative samples included in the analysis.

i) Supplementary Information:

In the Supplementary Information, the columns should be rearranged so that all of them fit on the same page, even if this requires changing the page orientation (e.g., to landscape format). In the current layout, the information is difficult to follow.

## AUTHORS' RESPONSE TO THE REVIEWERS

Point-by-Point Response to Comments

Thank you very much for taking the time to review this manuscript. Please find the detailed responses below, along with the corresponding revisions and corrections highlighted in track changes in the resubmitted files.

REVIEWER: 1

General Comments

The text would benefit from improved fluency by restructuring long sentences, incorporating clear connectors, and enhancing the logical flow of ideas.

R: Multiple changes were made throughout the text, many of which are detailed below, to improve the text and to comply with the reviewer's request. Other changes are indicated by the tracked changes in the manuscript.

It is also recommended that the discussion be expanded to better address the study's limitations, potential improvements to the test, and the practical implications for its field application.

R: Paragraphs were inserted in the discussion to improve explanations on the limitations and improvements that were observed in the study. Lines 440 – 453, 513 – 523, and 537 – 554.

Additionally, it is important to include confidence intervals for the reported sensitivity and specificity values, as these are essential for assessing the robustness and reliability of the results.

R: Thanks for the suggestion. The confidence intervals (CI) are included in the revised figures 2B and 3B, where we confirm that we used the 95% CI, as indicated in parentheses.

In the introduction, it is important to highlight the use of Protein A as a universal conjugate, emphasizing its applicability and performance with both human and canine IgG.

R: Thanks for the suggestion. For the revised manuscript, we added a paragraph in the introduction that explains the potential use of protein A (*Staphylococcus aureus*), which has high affinity for several classes of antibodies from various species, in assays for both humans and other animals. We have also included a new reference (Hober et al. 2007) with regard to the new text (New lines 127 – 132). In the new paragraph, we now also introduced our aim of finding an antigen that is efficient for both diagnostic tests and the previous work identifying antigens that were good for human or canine sera, but not both, as well as the early Q5 results  (New lines 127 – 133).

Include an explanation of the differences between Lateral Flow (LF) and Dual Path Platform (DPP).

R: To accommodate this requirement, we have also added a paragraph, but in the discussion, comparing the LF and DPP platforms, with corresponding new references (Gunasekera et al. 2015 and Silva et al. 2020), and explaining why we think that the LF approach might be more cost-effective (New lines 513 – 523).

It is also essential to underscore a key advantage: the standardized conditions allow the same rapid test to be used for diagnosing both humans and dogs in field settings. This approach streamlines production and improves cost-effectiveness.

R: Thanks for the suggestion, since it reinforces the relevance of our work. To consider that, we have included as the last sentence of the manuscript the text "the standardized conditions allowing the same rapid test to be used for diagnosing both humans and dogs in field settings should greatly facilitate the streamlined production of a single test for both hosts and improve its cost-effectiveness and widespread use" (New lines 551-554).

Additionally, the complementary role of ELISA and the rapid test (RDT) should be reinforced, as their combined use enhances the accuracy and reliability of Leishmania infantum infection detection.

R: Thanks for the suggestion. At the end of the introduction for the revised manuscript, last two sentences, we consider based on the reviewer's suggestion that "that combining the Q5-based RDT with the ELISA may further enhance diagnostic robustness" and also "may provide a complementary approach that improves accuracy, particularly in diverse clinical settings or in cases with borderline results" (New lines 153- 157).

It should be clearly stated which platform is intended for screening and which one is proposed as a confirmatory test, considering their suitability for large-scale surveillance in transmission areas.

R: In the last line of the introduction, also based on the reviewer's suggestion, we now specifically state that the most likely approach envisioned by us has the RDT as a screening assay with the ELISA having a confirmatory role  (New lines 155 – 157).

Include a table summarizing all the production conditions of the evaluated batches, along with the corresponding serum sample testing results.

R: Since the data from the ELISA and RDT assays are fundamentally different, we believe that joining them together in a single Table would not be possible. Currently, all major results and information regarding the assays are summarized in individual Tables within Figures 2, 3, 5, and 6, which we considered was the best way to display the results next to the relevant data (for the ELISA assays, for instance, next to the ROC curves) and as they were mentioned in the text. Unless this is strictly necessary, we would prefer to keep the data in the separate tables as they are.

Title: Evaluation of a new rapid diagnostic test based on the chimeric protein Q5 for the diagnosis of human and canine visceral leishmaniasis.

R: Thanks for the suggestion. The title has been improved, as suggested, to: "Evaluation of a new rapid diagnostic test, based on the chimeric protein Q5, for the diagnosis of human and canine forms of visceral leishmaniasis."

Keywords: RDT; ELISA; Visceral Leishmaniasis; recombinant protein.

R: Thanks for the suggestion. The Keywords have been modified as suggested. (New line 68).

Introduction (Lines 139-144)

In the present study, the recombinant Q5 protein was evaluated using a larger panel of human and canine sera through both ELISA and rapid diagnostic test (RDT). The Q5-based RDT demonstrated equivalent or superior sensitivity and specificity compared to current diagnostic tests for both human and canine visceral leishmaniasis (VL), reinforcing its potential as a reliable tool for VL diagnosis. Based on these findings, we hypothesize that combining the Q5-based RDT with the ELISA may further enhance diagnostic robustness, offering a complementary approach that improves accuracy, particularly in diverse clinical settings or in cases with borderline results.

R: Thanks for the suggestion. We have modified the text, replacing, as suggested, the previous text with the one above (New lines 147 – 157).

Fig. 1. Description of human and canine serum panels employed in this study: Geographic distribution in Brazil, highlighting the states where human (blue) and canine (green) serum samples were collected. These samples were used throughout the study, except for the VL-positive and VL-negative human sera from Minas Gerais, which were exclusively tested with the RDT-Q5 by FUNED-LACEN.

R: Thanks for the suggestion. We have replaced the figure legend with the new text (New lines 265 – 268).

Fig. 2. – provide a Y-axis label

R: Thanks for the suggestion. We have included Y-axis labels for both panels in Figure 2.

Results Section:

Please verify whether some content overlaps with the Materials and Methods or Discussion sections, as it appears there may be some mixing. Previously published results should be included only in the general discussion, not in the results section.

R: We have reviewed the text thoroughly but found that any minor overlaps with Materials and Methods were required for a better understanding of the work. Overlaps with the Discussion were reduced by transferring most of the comments on previous results (and references) to the discussion or removing them altogether (see new Lines

245-247). We felt, however, that it was necessary to keep some of the mentions to our previous work (Santos et al., 2020) to justify some of the procedures and analysis that were performed (in new line 247).

I recommend adding an introductory paragraph summarizing the main findings of this study, followed by a clear presentation of the results organized by specific topics. Including a summary table with consolidated data, absolute numbers (XX/XX), and corresponding percentages is also advisable.

R: The introductory paragraph summarizing the main findings is placed at the end of the introduction. We feel that further summarizing these at the beginning of the "Results" would only add redundancy and repeated text. Regarding the "summary table", we have discussed above the reasons why we thought that such a table would not adequately fit into the way our manuscript is currently structured.

Clarify the evaluation of the test with whole blood samples under real field conditions.

R: This has not been done yet, but in the revised discussion, we have raised this issue in the last paragraph of the discussion, by adding the following sentence: "Further validation of the test will require its assessment with whole blood samples under simulated or real field conditions, to confirm the test's applicability and its robustness in real-world settings." (New lines 548-550).

Discussion Section:

The discussion is generally well presented. However, it would benefit from an introductory paragraph summarizing the study's objectives and relevance before discussing the findings in detail.

R: Based on the reviewer's comment, we revised the first paragraph of the discussion and indeed found it too complex, despite already including some of the information asked by the reviewer. In the revised manuscript, we restricted the first paragraph to a summarized revision of our results, maintaining only relevant references related to that, some previously found in the "Results" section. Remaining topics of discussion were moved forward to a new paragraph immediately after (New lines 440 to 453).

It is recommended to include a specific evaluation of the test using whole blood under field conditions. This assessment is essential to demonstrate the test's applicability and robustness in real-world settings, beyond laboratory conditions. Providing this information will strengthen the evidence for the test's operational performance and its suitability for point-of-care use in both human and canine populations.

R: We haven't yet been able to evaluate the test under field conditions with blood samples, but we agree that this needs to be done in the future to confirm its applicability and usefulness in the field. As detailed above, in the revised manuscript, we have included an additional sentence in the last paragraph of the discussion to consider this limitation (New lines 550-554).

Further comments are provided directly in the manuscript file.

R: We haven't found or received any manuscript file with additional comments, so we believe that all of the reviewer's comments were the ones listed above and have replied accordingly to all.

REVIEWER: 2
c) Introduction
On page 6, the paragraph between lines 84–96 begins with the statement: "Clinical canine VL diagnosis can be impaired due to its presentation overlapping with other infections or hematological malignancies." However, the subsequent information applies to both canine and human VL, and most of the cited references pertain to human VL. Furthermore, in human VL, the clinical features are also not pathognomonic, as symptoms overlap with those of other diseases. The authors should revise this paragraph to improve clarity and ensure consistency.

R: Thanks for the suggestion. For a clearer understanding of the text and also more consistency, we decided to remove the sentence "*Clinical canine VL diagnosis can be impaired due to its presentation overlapping with other infections or hematological malignancies*" (see new lines 84 to 87).

d) Methodology
The authors did not describe the samples from Minas Gerais (MG) in the Materials and Methods section; instead, this information appears in Figure 1. The authors should include a description of these samples in the Materials and Methods section.

R: Thanks for the suggestion. For a better understanding of the text, we have included the following sentence: "To complete the studies, a last batch of sera was evaluated by the Enrique Dias Foundation (FUNED) from the state of Minas Gerais (48 positive and 52 negative sera) (Fig. 1A). (New lines 178 to 180).

Please place the number of the figure at the end of the sentence (line 178).

R: Thanks for the suggestion. Indeed, the description of the canine sera did not reference Figure 1, either in the Methodology or Results sections. In the revised manuscript, we have specifically mentioned the figure (Figure 1B) in both sections (new lines 184 and 282). We also added the labels "A" and "B" for the panels representing the human and canine sera, respectively.

The authors state in the Statistical Analysis section that Cohen's Kappa (κ) analysis was performed; however, the results of this analysis are not included in the manuscript. This should be addressed.

R: Thanks for the suggestion. The Kappa index was not actually used in this study. For this reason, the methodology that describes its use was taken from the "Statistical Analyses" section (see new lines 235 to 241).

e) Results

It is recommended that the authors specify Figure 1A in line 246.

R: Thanks for the suggestion. We have modified as suggested in new line 180. As stated above, we also added the labels "A" and "B" for the panels representing the human and canine sera, respectively.

On page 18, lines 353–355: "It was sent to an independent laboratory (FUNED-LACEN, from the state of Minas Gerais Q5/2021 (MG) where values of 98% sensitivity (N=48) and 90% specificity (N=52) were observed." — please close parentheses appropriately.

R: Thanks for pointing this out. We have fixed this in new lines 350 to 352.

The authors should review the results for batches Q5/2017 and Q5/2021 in Figure 6B (page 20) and revise the corresponding text in the results section (lines 383–394) for consistency.

R: Thanks again to the reviewer for pointing out the inconsistency. We have fixed this for both Q5/2017 (new lines 378 to 380) and Q5/2021 (new line 389).

The samples from Pernambuco, both human and canine, showed lower sensitivity compared to the other groups. Could the authors comment on whether this might be related to the diagnostic tests used? For dogs, serology was used; for humans, PCR and serology were applied in 22 of the 31 samples.

R: Indeed, the point raised by the reviewer is relevant considering the ELISA-Q5. For the canine sera, two of the thirteen sera tested from Pernambuco had false negative results, with those also having a negative PCR result, and we cannot rule out some impact from our use of other serological methods to confirm the VL diagnosis. Nevertheless, they represent only a minor fraction of the total number of sera tested and do not impact the overall evaluation of the test. For the human sera from Pernambuco, there is a greater incidence of false negative results for those sera tested only through sorology/PCR (five of 22 samples, with four of those having a positive PCR result), when compared with those parasitologically confirmed sera (one false negative among nine samples). But as for the other human sera, there is a clear increase in performance with the RDT-Q5 when compared with the ELISA-Q5, which is the main point we want to make. Therefore, unless it is deemed to be strictly required by the reviewer, we prefer not to raise this issue in the manuscript within the results section, since in our view it does not impact our conclusions and it may confuse the reader. As detailed below, however, we did include a sentence in the discussion to consider the possibility of a lower performance with the Pernambuco samples due to a lack of parasitological confirmation of some of the samples.

f) Discussion

In the first paragraph, the authors could enhance the discussion by addressing not only immunosuppression but also age as a key factor associated with the lower accuracy of serological tests for human visceral leishmaniasis (Freire et al., 2019; Cota et al., 2012).

R: In the revised manuscript, the discussion of the impact of immunosuppression on the VL diagnosis was moved to the 2nd paragraph, but we included age as a possible interfering effect (new lines 458 – 460). Related to that, we also included the reference by Freire et al., 2019.

The smaller number of negative controls in both human and canine samples, compared to the positive ones, should be acknowledged as a limitation of the study in the discussion.

R: A new sentence has been included in the discussion mentioning the need for a subsequent analysis with a greater number of negative sera. New lines 541 – 543.

It is recommended to include in the discussion the limitation of not incorporating samples from dogs with other diseases—such as trypanosomatids, Ehrlichia canis, Babesia canis, and other Leishmania species—among the truly negative control group.

R: Another sentence was included in the discussion, also considering the need for these samples in subsequent studies, New lines 545-546.

The discussion should also address the limitations of using a small number of control samples with potential cross-reactivity (e.g., Chagas disease and cutaneous leishmaniasis) and the lack of samples from humans with other diseases, such as paracoccidioidomycosis, malaria, and active pulmonary tuberculosis.

R: Yet another sentence was included to consider the human control samples. New lines 546 – 548.

Additionally, the discussion should mention the limitations of using serology rather than parasitological tests as the reference standard to classify the 13 dogs from Pernambuco as positive, and how this could introduce classification bias.

R: To comply with the reviewer's request, we have included the following sentence in the discussion "Subsequent studies will require as much as possible the use of sera from individuals with the VL diagnosis confirmed parasitologically, or at least through PCR, to avoid bias introduced by previous testing with other serological methods and which partially might be a reason for the reduced sensitivity observed here with the ELISA-Q5 for both human and canine samples from the State of Pernambuco." (New lines 537 – 541.)

h) Figures and tables

I suggest that the authors include a table with the demographic data of the human patients used in the study.

R: As in our previous manuscript (Santos et al., Plos NTD 2020), our main goal here was to use positive/negative samples to evaluate performance in VL diagnosis, with no intention to assess any further interfering circumstances from the human patients, which might impact diagnosis. Furthermore, the human sera assessed here were derived from different research groups and originally collected with different aims, which might impact the type of data available for individual patients. Based on these reasons, we considered that compiling such a table with standardized information for all patients might not be possible and would not add any relevant information regarding the goals we set out for this manuscript. Therefore, unless this is considered a strict requirement for publication, we respectfully opted not to include such a table in the manuscript.

Does Figure 1A refer to the "blue map"? The authors should ensure consistent labeling between the figure and the text (line 165).

R: Thanks for the suggestion. We have fixed the labels in Figure 1 for the revised manuscript, with A and B added to the figure, representing the two maps.

In Figure 2A, the Y-axis should be labeled as "Optical Density."

R: We have fixed the lack of a label for the Y-axis, as also indicated by the 1st reviewer, for Figure 2A.

In Figure 2C, the Y-axis should be labeled as "Sensitivity (%)". Additionally, please verify that the identification of the curves (HVL-MS and HVL-PE) matches that in Figures 2A and 2B.

R: We have also fixed the label for the Y-axis for Figure 2C. We have also fixed the identification of the two curves, which were mixed up. Thanks to the reviewer for pointing this out.

In Figure 2B, the summary of the ELISA-Q5 results should be presented similarly to how it is shown in Figure 3B. In both figures, the total number of positive samples should be included in the last line. Furthermore, the number of negative samples should be specified, as all figures and tables should be self-explanatory, without requiring the reader to refer back to the text.

R: Thanks for the suggestions. We have fixed the two Figures, 2B and 3B, which have equivalent data and labels now, and should be clearer.

Conceptually, positive and negative predictive values (PPV and NPV) depend on the prevalence of the disease in the population being tested. When samples are selected based on prior diagnosis, PPV and NPV will vary according to the proportion of positive and negative samples included in the analysis.

R: Thanks for the suggestions. Based on this, we prefer to remove the positive and negative predictive values (Figures 2B and 3B), keeping only the accuracy, as this characterizes the test performance.

i) Supplementary Information:

In the Supplementary Information, the columns should be rearranged so that all of them fit on the same page, even if this requires changing the page orientation (e.g., to landscape format). In the current layout, the information is difficult to follow.

## SECOND REVIEW ROUND

REVIEWERS' COMMENTS

### REVIEWER #1

Dear authors,

Congratulations on submitting your manuscript. While I have recommended its acceptance, some minor corrections need to be made:

1. When the rapid test is negative, only the control line appears, and if the control line does not appear, the test is considered invalid. Therefore, please replace "test line" with "control line" in lines 228 and 229.

Additionally, in the legend for Figure 4, in line 324, change "Control Line" to "Test Line" and "no reactivity" to "invalid test." In line 325, replace "Test Line" with "Control Line."

2. In the statistical analysis section, please remove "positive and negative predictive values" from line 234.

3. Also in the statistical analysis, in lines 239 and 240, please add the phrase "optical density." Specifically, include the following: "The cut-off was determined by adding twice the standard deviation of the negative sera optical density to the mean optical density of the negative samples, corresponding to a 95% confidence interval."

Best regards

**REVIEWER #2**

After reviewing the revised version of the manuscript IOC-2025-0126.R1, I am pleased to report that the authors have addressed all major concerns raised during the review process in a thorough and satisfactory manner. The revised submission demonstrates substantial improvement in clarity, organization, and scientific rigor. These modifications significantly enhance both the readability and interpretability of the study, making suitable for publication.

The manuscript exhibits scientific and technical merit, presenting a designed and methodologically evaluation of a chimeric antigen (Q5) for use in the serodiagnosis of visceral leishmaniasis in both humans and dogs. The inclusion of multiple independent serum panels adds robustness to the findings. The proposed Q5-based rapid test demonstrates excellent diagnostic performance and holds high translational potential, particularly for field applications.

Overall, the study represents an original and valuable contribution, providing important insights into the development of a dual diagnostic tool for human and canine visceral leishmaniasis. The methodological approach is appropriate, the results are statistically well-supported, and the work offers clear practical value—especially for improving field-level diagnosis and enhancing cost-effectiveness and public health programs.

### AUTHORS' RESPONSE TO THE REVIEWERS

Point-by-Point Response to Comments_V2

Thank you very much for taking the time to review this manuscript. Please find the detailed responses below, along with the corresponding revisions and corrections highlighted in track changes in the resubmitted files.

REVIEWER: 1
General Comments
1. When the rapid test is negative, only the control line appears, and if the control line does not appear, the test is considered invalid. Therefore, please replace "test line" with "control line" in lines 228 and 229. Additionally, in the legend for Figure 4, in line 324, change "Control Line" to "Test Line" and "no reactivity" to "invalid test." In line 325, replace "Test Line" with "Control Line."
R: Thanks for the suggestion. I have made the suggested corrections in the lines. The 228 and 229 lines, like the 324 and 325 lines.

2. In the statistical analysis section, please remove "positive and negative predictive values" from line 234.
R: Thanks for the suggestion. Remove the positive and negative predictive values. Now the new sentence is, "Sensitivity, specificity, and confidence interval parameters were estimated with the software MedCalc (version 12.3) (MedCalc Software, Ostend, Belgium)".

3. Also in the statistical analysis, in lines 239 and 240, please add the phrase "optical density." Specifically, include the following: "The cut-off was determined by adding twice the standard deviation of the negative sera optical density to the mean optical density of the negative samples, corresponding to a 95% confidence interval."
R: Thanks for the suggestion.  As suggested, the following text has been included in lines 239 and 240. The cut-off was determined by adding twice the standard deviation of the negative sera optical density to the mean optical density of the negative samples, corresponding to a 95% confidence interval.

REVIEWER: 2
Thank you for your comments and help in improving the understanding of the text.

## THIRD REVIEW ROUND

REVIEWERS' COMMENTS

**REVIEWER #1**

Dear Authors,
Thank you for making the corrections.
Kind regards,

**REVIEWER #2**

MIOC-2025-0126.R2
Lines 1-2: Evaluation of a Q5 Chimeric Protein in Rapid Diagnostic Test for Human and Canine Visceral Leishmaniasis

Comments in Methodology and results;
Line 204: Replace ELISA assays  with ELISA assays using recombinant proteins (rELISA)
Line 212: Replace RDT optimization with RDT based on the recombinant Q5
Line 230: Replace Multiple tests were then set up.... with Tests were then set up....
Line 247: Replace Reassessment of the ELISA-Q5 for the diagnosis of human VL with ELISA-Q5 for the diagnosis of human VL
Line 281: Replace Reassessment of the ELISA-Q5 for the canine VL diagnosis with ELISA-Q5 for the canine VL diagnosis
Line 309: Replace A new RDT based on the recombinant Q5 with RDT based on the recombinant Q5
Line 410: Replace A direct comparison of the Q5 performance using the ELISA and RDT tests with Comparative analysis of the diagnostic performance of Q5 using ELISA and RDT
Remove lines 248–250, as this information has already been presented in the Materials and Methods section (lines 160–164). "Our first analysis of the recombinant Q5 protein for the diagnosis of human VL using ELISA tested a total of 50 positive sera, all parasitologically confirmed from the Brazilian State of Piauí (PI), plus 50 VL-negative sera (Santos et al. 2020)."
Move Figure 1 to the Materials and Methods section, at the end of the sample description (line 193). Lines 253-254(the origins of all the sera used here and in the following topics are represented in Fig. 1A)
The cutoff value was not presented in the results for canine VL  and Human (Fig.2A).
Remove lines 282-284, as this information has already been presented in the Materials and Methods section "The previous analysis of the ELISA-Q5 for diagnosing canine VL evaluated 39 VL-positive sera from parasitologically confirmed dogs from Bahia, all of which produced positive results (Santos et al. 2020)."
Move lines 309–329 to the Materials and Methods section, after the RDT optimization subsection.
Lines 334-335: results shown in Fig. 5A. The RDT-Q5 was first evaluated with a limited panel of 40 VL-positive human sera from Piaui. Replace 40 with 41, as indicated in the Materials and Methods section.
Line 339–340 and Line 387: delete "different" and replace with "from three Brazilian States (Piauí, Mato Grosso do Sul, and Pernambuco)". .....sera from four Brazilian States, including those sera already assessed with the...
Line 341: replace "healthy" with "negative", so it reads "A total of 64 sera from negative controls". Lines 360 e 382: replace "healthy" with "negative" controls".
Lines 437–438: Replace with "Table 1. Agreement between the performance of ELISA-Q5 and RDT-Q5 using VL-positive and VL-negative human (HVL) and canine (CVL) sera."
Line 483: viable alternative for the VL diagnosis, with potential use for both forms of the disease. Replace both forms of the disease with for the diagnosis of human or canine VL.
Line 488: sera qualifies it as a cost-effective alternative for the diagnosis of human disease. Replace sera qualifies it as a cost-effective alternative for the diagnosis of human infection.

Comments in discussion: The occurrence of cross-reactivity with serum samples from dogs diagnosed with cutaneous leishmaniasis has been reported recently and represents a relevant challenge for the interpretation of serological tests. This finding may be associated with antigenic overlap among Leishmania species, particularly in light of isolated reports of American cutaneous leishmaniasis (ACL) caused by Leishmania (L.) infantum. These cases have been described in patients with or without HIV coinfection in the Central-West and Southwestern regions of Brazil (DOI: 10.1016/j.abd.2020.02.003). This epidemiological scenario reinforces the need for caution in interpreting serological results, as the atypical circulation of L. infantum in cutaneous presentations may contribute to cross-reactivity, affecting test specificity and highlighting the importance of combining diagnostic methods for accurate differentiation of leishmaniasis clinical forms.

Point-by-Point Response to Comments_V3

Thank you very much for taking the time to review this manuscript. Please find detailed responses below, along with the corresponding revisions and corrections highlighted in track changes in the resubmitted files.

REVIEWER: 2

General Comments

1. Line 204: Replace ELISA assays with ELISA assays using recombinant proteins (rELISA).

R: Thanks for the suggestion. I have made the suggested corrections in line 204: (*ELISA assays using recombinant proteins (rELISA).*

2. Line 212: Replace RDT optimization with RDT based on the recombinant Q5.

R: Thanks for the suggestion. I have made the suggested corrections in line 212: *RDT based on the recombinant.* I didn't specifically put Q5, because in this topic it was also done with other recombinants (Lci2 and Lci13).

3. Line 230: Replace Multiple tests were then set up.... with Tests were then set up....

R: Thanks for the suggestion. I have made the suggested corrections in line 230.

4. Line 247: Replace Reassessment of the ELISA-Q5 for the diagnosis of human VL with ELISA-Q5 for the diagnosis of human VL.

R: Thanks for the suggestion. I have made the suggested corrections in new line 243: *ELISA-Q5 for the diagnosis of human VL.*

5. Line 281: Replace Reassessment of the ELISA-Q5 for the canine VL diagnosis with ELISA-Q5 for the canine VL diagnosis.

R: Thanks for the suggestion. I have made the suggested corrections in new line 278: *ELISA-Q5 for the canine VL diagnosis.*

6. Line 309: Replace A new RDT based on the recombinant Q5 with RDT based on the recombinant Q5.

R: Thanks for the suggestion. I have made the suggested corrections in new line 307: *RDT based on the recombinant Q5.*

7. Line 410: Replace A direct comparison of the Q5 performance using the ELISA and RDT tests with Comparative analysis of the diagnostic performance of Q5 using ELISA and RDT.

R: Thanks for the suggestion. I have made the suggested corrections in new line 408: *Comparative analysis of the diagnostic performance of Q5 using ELISA and RDT.*

8. Remove lines 248–250, as this information has already been presented in the Materials and Methods section (lines 160–164). "Our first analysis of the recombinant Q5 protein for the diagnosis of human VL using ELISA tested a total of 50 positive sera, all parasitologically confirmed from the Brazilian State of Piauí (PI), plus 50 VL-negative sera (Santos et al. 2020).

R: Thanks for the suggestion. Remove the "Our first analysis of the recombinant Q5 protein for the diagnosis of human VL using ELISA tested a total of 50 positive sera, all parasitologically confirmed from the Brazilian State of Piauí (PI), plus 50 VL-negative sera (Santos et al. 2020)". *Now the new sentence is, "Thirty-one sera from Pernambuco were first tested with six producing false negative results and defining a sensitivity of 81%. Another 30 sera from Mato Grosso do Sul were also tested, with three false negative results and 90% sensitivity. To determine specificity, 64 VL-negative sera were also tested, with only one producing a false positive result (Fig. 2A and 2B)."* Lines 252 – 257.

9. Move Figure 1 to the Materials and Methods section, at the end of the sample description (line 193). Lines 253-254 (the origins of all the sera used here and in the following topics are represented in Fig. 1A).

R: Thanks for the suggestion. As suggested, the following text has been included in new lines 192 and 193 (the origins of all the sera used here and in the following topics are represented in Fig. 1A). As suggested, move Figure 1 to the Materials and Methods section (Line 195).

10. The cutoff value was not presented in the results for canine VL and Human (Fig.2A).

R: Thanks for the suggestion. As suggested, the following text has been included in new lines 257 and 258 cutoff value Human (Another 30 sera from Mato Grosso do Sul were also tested, with three false negative results and 90% sensitivity (cutoff 0.089). 300 cutoff value Human, to determine specificity, we also tested 47 sera with a previously defined negative diagnosis, of which only one was a false positive (cutoff 0.286).

11. Remove lines 282-284, as this information has already been presented in the Materials and Methods section "The previous analysis of the ELISA-Q5 for diagnosing canine VL evaluated 39 VL-positive sera from parasitologically confirmed dogs from Bahia, all of which produced positive results (Santos et al. 2020)."

R: Thanks for the suggestion. As suggested, the following text has been removing the line *"The previous analysis of the ELISA-Q5 for diagnosing canine VL evaluated 39 VL-positive sera from parasitologically confirmed dogs from Bahia, all of which produced positive results (Santos et al. 2020)." Now the new sentence is, "In this study, we reassessed the ELISA-Q5 using a significantly larger sample of VL-positive canine sera (summarized in Fig. 1B)."*

12. Move lines 309–329 to the Materials and Methods section, after the RDT optimization subsection.

R: Thanks for the suggestion. We thank the reviewer for the suggestion. However, we chose to keep lines 309–329 in the Results section because the manuscript is primarily focused on the description of a new methodology (RDT-Q5). In this context, these data represent experimental outcomes obtained during the RDT optimization process rather than purely procedural steps. Therefore, we believe their placement in the Results section better reflects their role in demonstrating the performance and development of the assay.

13. Lines 334-335: results shown in Fig. 5A. The RDT-Q5 was first evaluated with a limited panel of 40 VL-positive human sera from Piaui. Replace 40 with 41, as indicated in the Materials and Methods section.

R: Thanks for the suggestion. I have made the suggested corrections in line 339.

14. Line 339–340 and Line 387: delete "different" and replace with "from three Brazilian States (Piauí, Mato Grosso do Sul, and Pernambuco)". ....sera from four Brazilian States, including those sera already assessed with the....

R: Thanks for the suggestion. I have made the suggested corrections in lines 345 and 392.

15. Line 341: replace "healthy" with "negative", so it reads "A total of 64 sera from negative controls". Lines 360 e 382: replace "healthy" with "negative" controls".

R: Thanks for the suggestion. I have made the suggested corrections in lines 346, 365 and 387.

16. Lines 437–438: Replace with "Table 1. Agreement between the performance of ELISA-Q5 and RDT-Q5 using VL-positive and VL-negative human (HVL) and canine (CVL) sera."

R: Thanks for the suggestion. I have made the suggested corrections in new line 443: *Table 1. Agreement between the performance of ELISA-Q5 and RDT-Q5 using VL-positive and VL-negative human (HVL) and canine (CVL) sera.*

17. Line 483: viable alternative for the VL diagnosis, with potential use for both forms of the disease. Replace both forms of the disease with for the diagnosis of human or canine VL.

R: Thanks for the suggestion. I have made the suggested corrections in new line 490: *both forms of the disease with for the diagnosis of human or canine VL.*

18. Line 488: sera qualifies it as a cost-effective alternative for the diagnosis of human disease. Replace sera qualifies it as a cost-effective alternative for the diagnosis of human infection.

R: Thanks for the suggestion. I have made the suggested corrections in new line 496: *sera qualify it as a cost-effective alternative for the diagnosis of human infection.*

**REVIEWER #2**

No further comments. The authors have adequately addressed all the points raised.

