## [Reviewer Report · FIRST REVIEW ROUND - REVIEWERS COMMENTS]

## REVIEWER #1

General Comments

I would like to thank you for the opportunity to review this manuscript. The approach, focused on achieving accurate diagnosis, is highly relevant for the control of human and canine visceral leishmaniasis, particularly within the context of public health. The manuscript presents solid scientific merit, with well-defined results and a clear contribution to improving the diagnosis of leishmaniasis. However, the text would benefit from improved fluency by restructuring long sentences, incorporating clear connectors, and enhancing the logical flow of ideas. It is also recommended that the discussion be expanded to better address the study’s limitations, potential improvements to the test, and the practical implications for its field application. Additionally, it is important to include confidence intervals for the reported sensitivity and specificity values, as these are essential for assessing the robustness and reliability of the results.

In the introduction, it is important to highlight the use of Protein A as a universal conjugate, emphasizing its applicability and performance with both human and canine IgG. Include an explanation of the differences between Lateral Flow (LF) and Dual Path Platform (DPP). It is also essential to underscore a key advantage: the standardized conditions allow the same rapid test to be used for diagnosing both humans and dogs in field settings. This approach streamlines production and improves cost-effectiveness. Additionally, the complementary role of ELISA and the rapid test (RDT) should be reinforced, as their combined use enhances the accuracy and reliability of *Leishmania infantum* infection detection. It should be clearly stated which platform is intended for screening and which one is proposed as a confirmatory test, considering their suitability for large-scale surveillance in transmission areas. Include a table summarizing all the production conditions of the evaluated batches, along with the corresponding serum sample testing results.

Title: Evaluation of a new rapid diagnostic test based on the chimeric protein Q5 for the diagnosis of human and canine visceral leishmaniasis

Keywords: RDT; ELISA; Visceral Leishmaniasis; recombinant protein.

Introduction (Lines 139-144)

In the present study, the recombinant Q5 protein was evaluated using a larger panel of human and canine sera through both ELISA and rapid diagnostic test (RDT). The Q5-based RDT demonstrated equivalent or superior sensitivity and specificity compared to current diagnostic tests for both human and canine visceral leishmaniasis (VL), reinforcing its potential as a reliable tool for VL diagnosis. Based on these findings, we hypothesize that combining the Q5-based RDT with the ELISA may further enhance diagnostic robustness, offering a complementary approach that improves accuracy, particularly in diverse clinical settings or in cases with borderline results.

Fig. 1. Description of human and canine serum panels employed in this study: Geographic distribution in Brazil highlighting the states where human (blue) and canine (green) serum samples were collected. These samples were used throughout the study, except for the VL-positive and VL-negative human sera from Minas Gerais, which were exclusively tested with the RDT-Q5 by FUNED-LACEN.

Fig. 2. – provide an Y-axis label

Results Section:

Please verify whether some content overlaps with the Materials and Methods or Discussion sections, as it appears there may be some mixing. Previously published results should be included only in the general discussion, not in the results section. I recommend adding an introductory paragraph summarizing the main findings of this study, followed by a clear presentation of the results organized by specific topics. Including a summary table with consolidated data, absolute numbers (XX/XX) and corresponding percentages, is also advisable. Clarify the evaluation of the test with whole blood samples under real field conditions.

Discussion Section:

The discussion is generally well presented. However, it would benefit from an introductory paragraph summarizing the study’s objectives and relevance before discussing the findings in detail. It is recommended to include a specific evaluation of the test using whole blood under field conditions. This assessment is essential to demonstrate the test’s applicability and robustness in real-world settings, beyond laboratory conditions. Providing this information will strengthen the evidence for the test’s operational performance and its suitability for point-of-care use in both human and canine populations.

Further comments are provided directly in the manuscript file.

Sincerely,

## REVIEWER #2

Evaluation of a new rapid diagnostic test based on the chimeric protein Q5 and applicable for the diagnosis of human and canine forms of visceral leishmaniasis

The manuscript by dos Santos et al. presents a reassessment of the Q5 chimeric recombinant protein using the ELISA platform with a larger set of human and canine VL-positive sera. It also describes the development of a new RDT-Q5, tested with human and canine sera from various states in Brazil. The study is scientifically sound, and the results indicate that Q5 shows promise for the serological diagnosis of VL—through ELISA assays for dogs and RDTs for humans. The article is well-written, and the results adequately support the conclusions. However, the authors must address sime issues.

a) Adequacy of the abstract

The abstract is well-written and provides a clear summary of the study.

b) Originality and importance of the contribution to the field

The study is both relevant and innovative, as it aims to demonstrate the utility of the Q5 chimeric protein on an immunochromatographic platform for the diagnosis of human and canine leishmaniasis.

c) Introduction

On page 6, the paragraph between lines 84–96 begins with the statement: “Clinical canine VL diagnosis can be impaired due to its presentation overlapping with other infections or hematological malignancies.” However, the subsequent information applies to both canine and human VL, and most of the cited references pertain to human VL. Furthermore, in human VL, the clinical features are also not pathognomonic, as symptoms overlap with those of other diseases. The authors should revise this paragraph to improve clarity and ensure consistency.

d) Methodology

The methodology is well described. Specific comments: The authors did not describe the samples from Minas Gerais (MG) in the Materials and Methods section; instead, this information appears in Figure 1. The authors should include a description of these samples in the Materials and Methods section. Please place the number of the figure at the end of the sentence (line 178). The authors state in the Statistical Analysis section that Cohen’s Kappa (κ) analysis was performed; however, the results of this analysis are not included in the manuscript. This should be addressed.

e) Results

The results are well presented. However, the following points could help improve the presentation: It is recommended that the authors specify Figure 1A in line 246. On page 18, lines 353–355: “It was sent to an independent laboratory (FUNED-LACEN, from the state of Minas Gerais Q5/2021 (MG) where values of 98% sensitivity (N=48) and 90% specificity (N=52) were observed.” — please close parentheses appropriately. The authors should review the results for batches Q5/2017 and Q5/2021 in Figure 6B (page 20) and revise the corresponding text in the results section (lines 383–394) for consistency. The samples from Pernambuco, both human and canine, showed lower sensitivity compared to the other groups. Could the authors comment on whether this might be related to the diagnostic tests used? For dogs, serology was used; for humans, PCR and serology were applied in 22 of the 31 samples.

f) Discussion

In the first paragraph, the authors could enhance the discussion by addressing not only immunosuppression but also age as a key factor associated with the lower accuracy of serological tests for human visceral leishmaniasis (Freire et al., 2019; Cota et al., 2012). The smaller number of negative controls in both human and canine samples, compared to the positive ones, should be acknowledged as a limitation of the study in the discussion. It is recommended to include in the discussion the limitation of not incorporating samples from dogs with other diseases—such as trypanosomatids, *Ehrlichia canis*, *Babesia canis*, and other *Leishmania* species—among the truly negative control group. The discussion should also address the limitation of using a small number of control samples with potential cross-reactivity (e.g., Chagas disease and cutaneous leishmaniasis) and the lack of samples from humans with other diseases, such as paracoccidioidomycosis, malaria, and active pulmonary tuberculosis. Additionally, the discussion should mention the limitation of using serology rather than parasitological tests as the reference standard to classify the 13 dogs from Pernambuco as positive, and how this could introduce classification bias.

g) References

Current, extensive and strategic references to the theme have been included.

h) Figures and tables

I suggest that the authors include a table with the demographic data of the human patients used in the study. Does Figure 1A refer to the “blue map”? The authors should ensure consistent labeling between the figure and the text (line 165). In Figure 2A, the Y-axis should be labeled as “Optical Density.” In Figure 2C, the Y-axis should be labeled as “Sensitivity (%)”. Additionally, please verify that the identification of the curves (HVL-MS and HVL-PE) matches those in Figures 2A and 2B. In Figure 2B, the summary of the ELISA-Q5 results should be presented similarly to how it is shown in Figure 3B. In both figures, the total number of positive samples should be included in the last line. Furthermore, the number of negative samples should be specified, as all figures and tables should be self-explanatory, without requiring the reader to refer back to the text. Conceptually, positive and negative predictive values (PPV and NPV) depend on the prevalence of the disease in the population being tested. When samples are selected based on prior diagnosis, PPV and NPV will vary according to the proportion of positive and negative samples included in the analysis.

i) Supplementary Information:

In the Supplementary Information, the columns should be rearranged so that all of them fit on the same page, even if this requires changing the page orientation (e.g., to landscape format). In the current layout, the information is difficult to follow.

## AUTHORS’ RESPONSE TO THE REVIEWERS

Point-by-Point Response to Comments

Thank you very much for taking the time to review this manuscript. Please find the detailed responses below, along with the corresponding revisions and corrections highlighted in track changes in the resubmitted files.

REVIEWER: 1

General Comments

The text would benefit from improved fluency by restructuring long sentences, incorporating clear connectors, and enhancing the logical flow of ideas.

R: Multiple changes were made throughout the text, many of which are detailed below, to improve the text and to comply with the reviewer’s request. Other changes are indicated by the tracked changes in the manuscript.

It is also recommended that the discussion be expanded to better address the study’s limitations, potential improvements to the test, and the practical implications for its field application.

R: Paragraphs were inserted in the discussion to improve explanations on the limitations and improvements that were observed in the study. Lines 440 – 453, 513 – 523, and 537 – 554.

Additionally, it is important to include confidence intervals for the reported sensitivity and specificity values, as these are essential for assessing the robustness and reliability of the results.

R: Thanks for the suggestion. The confidence intervals (CI) are included in the revised figures 2B and 3B, where we confirm that we used the 95% CI, as indicated in parentheses.

In the introduction, it is important to highlight the use of Protein A as a universal conjugate, emphasizing its applicability and performance with both human and canine IgG.

R: Thanks for the suggestion. For the revised manuscript, we added a paragraph in the introduction that explains the potential use of protein A (*Staphylococcus aureus*), which has high affinity for several classes of antibodies from various species, in assays for both humans and other animals. We have also included a new reference (Hober et al. 2007) with regard to the new text (New lines 127 – 132). In the new paragraph, we now also introduced our aim of finding an antigen that is efficient for both diagnostic tests and the previous work identifying antigens that were good for human or canine sera, but not both, as well as the early Q5 results (New lines 127 – 133).

Include an explanation of the differences between Lateral Flow (LF) and Dual Path Platform (DPP).

R: To accommodate this requirement, we have also added a paragraph, but in the discussion, comparing the LF and DPP platforms, with corresponding new references (Gunasekera et al. 2015 and Silva et al. 2020), and explaining why we think that the LF approach might be more cost-effective (New lines 513 – 523).

It is also essential to underscore a key advantage: the standardized conditions allow the same rapid test to be used for diagnosing both humans and dogs in field settings. This approach streamlines production and improves cost-effectiveness.

R: Thanks for the suggestion, since it reinforces the relevance of our work. To consider that, we have included as the last sentence of the manuscript the text “the standardized conditions allowing the same rapid test to be used for diagnosing both humans and dogs in field settings should greatly facilitate the streamlined production of a single test for both hosts and improve its cost-effectiveness and widespread use” (New lines 551-554).

Additionally, the complementary role of ELISA and the rapid test (RDT) should be reinforced, as their combined use enhances the accuracy and reliability of *Leishmania infantum* infection detection.

R: Thanks for the suggestion. At the end of the introduction for the revised manuscript, last two sentences, we consider based on the reviewer’s suggestion that “that combining the Q5-based RDT with the ELISA may further enhance diagnostic robustness” and also “may provide a complementary approach that improves accuracy, particularly in diverse clinical settings or in cases with borderline results” (New lines 153- 157).

It should be clearly stated which platform is intended for screening and which one is proposed as a confirmatory test, considering their suitability for large-scale surveillance in transmission areas.

R: In the last line of the introduction, also based on the reviewer’s suggestion, we now specifically state that the most likely approach envisioned by us has the RDT as a screening assay with the ELISA having a confirmatory role (New lines 155 – 157).

Include a table summarizing all the production conditions of the evaluated batches, along with the corresponding serum sample testing results.

R: Since the data from the ELISA and RDT assays are fundamentally different, we believe that joining them together in a single Table would not be possible. Currently, all major results and information regarding the assays are summarized in individual Tables within Figures 2, 3, 5, and 6, which we considered was the best way to display the results next to the relevant data (for the ELISA assays, for instance, next to the ROC curves) and as they were mentioned in the text. Unless this is strictly necessary, we would prefer to keep the data in the separate tables as they are.

Title: Evaluation of a new rapid diagnostic test based on the chimeric protein Q5 for the diagnosis of human and canine visceral leishmaniasis.

R: Thanks for the suggestion. The title has been improved, as suggested, to: “Evaluation of a new rapid diagnostic test, based on the chimeric protein Q5, for the diagnosis of human and canine forms of visceral leishmaniasis.”

Keywords: RDT; ELISA; Visceral Leishmaniasis; recombinant protein.

R: Thanks for the suggestion. The Keywords have been modified as suggested. (New line 68).

Introduction (Lines 139-144)

In the present study, the recombinant Q5 protein was evaluated using a larger panel of human and canine sera through both ELISA and rapid diagnostic test (RDT). The Q5-based RDT demonstrated equivalent or superior sensitivity and specificity compared to current diagnostic tests for both human and canine visceral leishmaniasis (VL), reinforcing its potential as a reliable tool for VL diagnosis. Based on these findings, we hypothesize that combining the Q5-based RDT with the ELISA may further enhance diagnostic robustness, offering a complementary approach that improves accuracy, particularly in diverse clinical settings or in cases with borderline results.

R: Thanks for the suggestion. We have modified the text, replacing, as suggested, the previous text with the one above (New lines 147 – 157).

Fig. 1. Description of human and canine serum panels employed in this study: Geographic distribution in Brazil, highlighting the states where human (blue) and canine (green) serum samples were collected. These samples were used throughout the study, except for the VL-positive and VL-negative human sera from Minas Gerais, which were exclusively tested with the RDT-Q5 by FUNED-LACEN.

R: Thanks for the suggestion. We have replaced the figure legend with the new text (New lines 265 – 268).

Fig. 2. – provide a Y-axis label

R: Thanks for the suggestion. We have included Y-axis labels for both panels in Figure 2.

Results Section:

Please verify whether some content overlaps with the Materials and Methods or Discussion sections, as it appears there may be some mixing. Previously published results should be included only in the general discussion, not in the results section.

R: We have reviewed the text thoroughly but found that any minor overlaps with Materials and Methods were required for a better understanding of the work. Overlaps with the Discussion were reduced by transferring most of the comments on previous results (and references) to the discussion or removing them altogether (see new Lines 245-247). We felt, however, that it was necessary to keep some of the mentions to our previous work (Santos et al., 2020) to justify some of the procedures and analysis that were performed (in new line 247).

I recommend adding an introductory paragraph summarizing the main findings of this study, followed by a clear presentation of the results organized by specific topics. Including a summary table with consolidated data, absolute numbers (XX/XX), and corresponding percentages is also advisable.

R: The introductory paragraph summarizing the main findings is placed at the end of the introduction. We feel that further summarizing these at the beginning of the “Results” would only add redundancy and repeated text. Regarding the “summary table”, we have discussed above the reasons why we thought that such a table would not adequately fit into the way our manuscript is currently structured.

Clarify the evaluation of the test with whole blood samples under real field conditions.

R: This has not been done yet, but in the revised discussion, we have raised this issue in the last paragraph of the discussion, by adding the following sentence: “Further validation of the test will require its assessment with whole blood samples under simulated or real field conditions, to confirm the test’s applicability and its robustness in real-world settings.” (New lines 548-550).

Discussion Section:

The discussion is generally well presented. However, it would benefit from an introductory paragraph summarizing the study’s objectives and relevance before discussing the findings in detail.

R: Based on the reviewer’s comment, we revised the first paragraph of the discussion and indeed found it too complex, despite already including some of the information asked by the reviewer. In the revised manuscript, we restricted the first paragraph to a summarized revision of our results, maintaining only relevant references related to that, some previously found in the “Results” section. Remaining topics of discussion were moved forward to a new paragraph immediately after (New lines 440 to 453).

It is recommended to include a specific evaluation of the test using whole blood under field conditions. This assessment is essential to demonstrate the test’s applicability and robustness in real-world settings, beyond laboratory conditions. Providing this information will strengthen the evidence for the test’s operational performance and its suitability for point-of-care use in both human and canine populations.

R: We haven’t yet been able to evaluate the test under field conditions with blood samples, but we agree that this needs to be done in the future to confirm its applicability and usefulness in the field. As detailed above, in the revised manuscript, we have included an additional sentence in the last paragraph of the discussion to consider this limitation (New lines 550-554).

Further comments are provided directly in the manuscript file.

R: We haven’t found or received any manuscript file with additional comments, so we believe that all of the reviewer’s comments were the ones listed above and have replied accordingly to all.

REVIEWER: 2

c) Introduction

On page 6, the paragraph between lines 84–96 begins with the statement: “Clinical canine VL diagnosis can be impaired due to its presentation overlapping with other infections or hematological malignancies.” However, the subsequent information applies to both canine and human VL, and most of the cited references pertain to human VL. Furthermore, in human VL, the clinical features are also not pathognomonic, as symptoms overlap with those of other diseases. The authors should revise this paragraph to improve clarity and ensure consistency.

R: Thanks for the suggestion. For a clearer understanding of the text and also more consistency, we decided to remove the sentence “Clinical canine VL diagnosis can be impaired due to its presentation overlapping with other infections or hematological malignancies” (see new lines 84 to 87).

d) Methodology

The authors did not describe the samples from Minas Gerais (MG) in the Materials and Methods section; instead, this information appears in Figure 1. The authors should include a description of these samples in the Materials and Methods section.

R: Thanks for the suggestion. For a better understanding of the text, we have included the following sentence: “To complete the studies, a last batch of sera was evaluated by the Enrique Dias Foundation (FUNED) from the state of Minas Gerais (48 positive and 52 negative sera) (Fig. 1A). (New lines 178 to 180).

Please place the number of the figure at the end of the sentence (line 178).

R: Thanks for the suggestion. Indeed, the description of the canine sera did not reference Figure 1, either in the Methodology or Results sections. In the revised manuscript, we have specifically mentioned the figure (Figure 1B) in both sections (new lines 184 and 282). We also added the labels “A” and “B” for the panels representing the human and canine sera, respectively.

The authors state in the Statistical Analysis section that Cohen’s Kappa (κ) analysis was performed; however, the results of this analysis are not included in the manuscript. This should be addressed.

R: Thanks for the suggestion. The Kappa index was not actually used in this study. For this reason, the methodology that describes its use was taken from the “Statistical Analyses” section (see new lines 235 to 241).

e) Results

It is recommended that the authors specify Figure 1A in line 246.

R: Thanks for the suggestion. We have modified as suggested in new line 180. As stated above, we also added the labels “A” and “B” for the panels representing the human and canine sera, respectively.

On page 18, lines 353–355: “It was sent to an independent laboratory (FUNED-LACEN, from the state of Minas Gerais Q5/2021 (MG) where values of 98% sensitivity (N=48) and 90% specificity (N=52) were observed.” — please close parentheses appropriately.

R: Thanks for pointing this out. We have fixed this in new lines 350 to 352.

The authors should review the results for batches Q5/2017 and Q5/2021 in Figure 6B (page 20) and revise the corresponding text in the results section (lines 383–394) for consistency.

R: Thanks again to the reviewer for pointing out the inconsistency. We have fixed this for both Q5/2017 (new lines 378 to 380) and Q5/2021 (new line 389).

The samples from Pernambuco, both human and canine, showed lower sensitivity compared to the other groups. Could the authors comment on whether this might be related to the diagnostic tests used? For dogs, serology was used; for humans, PCR and serology were applied in 22 of the 31 samples.

R: Indeed, the point raised by the reviewer is relevant considering the ELISA-Q5. For the canine sera, two of the thirteen sera tested from Pernambuco had false negative results, with those also having a negative PCR result, and we cannot rule out some impact from our use of other serological methods to confirm the VL diagnosis. Nevertheless, they represent only a minor fraction of the total number of sera tested and do not impact the overall evaluation of the test. For the human sera from Pernambuco, there is a greater incidence of false negative results for those sera tested only through sorology/PCR (five of 22 samples, with four of those having a positive PCR result), when compared with those parasitologically confirmed sera (one false negative among nine samples). But as for the other human sera, there is a clear increase in performance with the RDT-Q5 when compared with the ELISA-Q5, which is the main point we want to make. Therefore, unless it is deemed to be strictly required by the reviewer, we prefer not to raise this issue in the manuscript within the results section, since in our view it does not impact our conclusions and it may confuse the reader. As detailed below, however, we did include a sentence in the discussion to consider the possibility of a lower performance with the Pernambuco samples due to a lack of parasitological confirmation of some of the samples.

f) Discussion

In the first paragraph, the authors could enhance the discussion by addressing not only immunosuppression but also age as a key factor associated with the lower accuracy of serological tests for human visceral leishmaniasis (Freire et al., 2019; Cota et al., 2012).

R: In the revised manuscript, the discussion of the impact of immunosuppression on the VL diagnosis was moved to the 2nd paragraph, but we included age as a possible interfering effect (new lines 458 – 460). Related to that, we also included the reference by Freire et al., 2019.

The smaller number of negative controls in both human and canine samples, compared to the positive ones, should be acknowledged as a limitation of the study in the discussion.

R: A new sentence has been included in the discussion mentioning the need for a subsequent analysis with a greater number of negative sera. New lines 541 – 543.

It is recommended to include in the discussion the limitation of not incorporating samples from dogs with other diseases—such as trypanosomatids, *Ehrlichia canis*, *Babesia canis*, and other *Leishmania* species—among the truly negative control group.

R: Another sentence was included in the discussion, also considering the need for these samples in subsequent studies, New lines 545-546.

The discussion should also address the limitations of using a small number of control samples with potential cross-reactivity (e.g., Chagas disease and cutaneous leishmaniasis) and the lack of samples from humans with other diseases, such as paracoccidioidomycosis, malaria, and active pulmonary tuberculosis.

R: Yet another sentence was included to consider the human control samples. New lines 546 – 548.

Additionally, the discussion should mention the limitations of using serology rather than parasitological tests as the reference standard to classify the 13 dogs from Pernambuco as positive, and how this could introduce classification bias.

R: To comply with the reviewer’s request, we have included the following sentence in the discussion “Subsequent studies will require as much as possible the use of sera from individuals with the VL diagnosis confirmed parasitologically, or at least through PCR, to avoid bias introduced by previous testing with other serological methods and which partially might be a reason for the reduced sensitivity observed here with the ELISA-Q5 for both human and canine samples from the State of Pernambuco.” (New lines 537 – 541.)

h) Figures and tables

I suggest that the authors include a table with the demographic data of the human patients used in the study.

R: As in our previous manuscript (Santos et al., Plos NTD 2020), our main goal here was to use positive/negative samples to evaluate performance in VL diagnosis, with no intention to assess any further interfering circumstances from the human patients, which might impact diagnosis. Furthermore, the human sera assessed here were derived from different research groups and originally collected with different aims, which might impact the type of data available for individual patients. Based on these reasons, we considered that compiling such a table with standardized information for all patients might not be possible and would not add any relevant information regarding the goals we set out for this manuscript. Therefore, unless this is considered a strict requirement for publication, we respectfully opted not to include such a table in the manuscript.

Does Figure 1A refer to the “blue map”? The authors should ensure consistent labeling between the figure and the text (line 165).

R: Thanks for the suggestion. We have fixed the labels in Figure 1 for the revised manuscript, with A and B added to the figure, representing the two maps.

In Figure 2A, the Y-axis should be labeled as “Optical Density.”

R: We have fixed the lack of a label for the Y-axis, as also indicated by the 1st reviewer, for Figure 2A.

In Figure 2C, the Y-axis should be labeled as “Sensitivity (%)”. Additionally, please verify that the identification of the curves (HVL-MS and HVL-PE) matches that in Figures 2A and 2B.

R: We have also fixed the label for the Y-axis for Figure 2C. We have also fixed the identification of the two curves, which were mixed up. Thanks to the reviewer for pointing this out.

In Figure 2B, the summary of the ELISA-Q5 results should be presented similarly to how it is shown in Figure 3B. In both figures, the total number of positive samples should be included in the last line. Furthermore, the number of negative samples should be specified, as all figures and tables should be self-explanatory, without requiring the reader to refer back to the text.

R: Thanks for the suggestions. We have fixed the two Figures, 2B and 3B, which have equivalent data and labels now, and should be clearer.

Conceptually, positive and negative predictive values (PPV and NPV) depend on the prevalence of the disease in the population being tested. When samples are selected based on prior diagnosis, PPV and NPV will vary according to the proportion of positive and negative samples included in the analysis.

R: Thanks for the suggestions. Based on this, we prefer to remove the positive and negative predictive values (Figures 2B and 3B), keeping only the accuracy, as this characterizes the test performance.

i) Supplementary Information:

In the Supplementary Information, the columns should be rearranged so that all of them fit on the same page, even if this requires changing the page orientation (e.g., to landscape format). In the current layout, the information is difficult to follow.

---

## [Reviewer Report · REVIEWERS COMMENTS]

## REVIEWER #1

Dear authors,

Congratulations on submitting your manuscript. While I have recommended its acceptance, some minor corrections need to be made:

1. When the rapid test is negative, only the control line appears, and if the control line does not appear, the test is considered invalid. Therefore, please replace “test line” with “control line” in lines 228 and 229. Additionally, in the legend for Figure 4, in line 324, change “Control Line” to “Test Line” and “no reactivity” to “invalid test.” In line 325, replace “Test Line” with “Control Line.”

2. In the statistical analysis section, please remove “positive and negative predictive values” from line 234.

3. Also in the statistical analysis, in lines 239 and 240, please add the phrase “optical density.” Specifically, include the following: “The cut-off was determined by adding twice the standard deviation of the negative sera optical density to the mean optical density of the negative samples, corresponding to a 95% confidence interval.”

Best regards

## REVIEWER #2

After reviewing the revised version of the manuscript IOC-2025-0126.R1, I am pleased to report that the authors have addressed all major concerns raised during the review process in a thorough and satisfactory manner. The revised submission demonstrates substantial improvement in clarity, organization, and scientific rigor. These modifications significantly enhance both the readability and interpretability of the study, making suitable for publication. The manuscript exhibits scientific and technical merit, presenting a designed and methodologically evaluation of a chimeric antigen (Q5) for use in the serodiagnosis of visceral leishmaniasis in both humans and dogs. The inclusion of multiple independent serum panels adds robustness to the findings. The proposed Q5-based rapid test demonstrates excellent diagnostic performance and holds high translational potential, particularly for field applications. Overall, the study represents an original and valuable contribution, providing important insights into the development of a dual diagnostic tool for human and canine visceral leishmaniasis. The methodological approach is appropriate, the results are statistically well-supported, and the work offers clear practical value—especially for improving field-level diagnosis and enhancing cost-effectiveness and public health programs.

## AUTHORS’ RESPONSE TO THE REVIEWERS

Point-by-Point Response to Comments_V2

Thank you very much for taking the time to review this manuscript. Please find the detailed responses below, along with the corresponding revisions and corrections highlighted in track changes in the resubmitted files.

REVIEWER: 1

General Comments

1. When the rapid test is negative, only the control line appears, and if the control line does not appear, the test is considered invalid. Therefore, please replace “test line” with “control line” in lines 228 and 229. Additionally, in the legend for Figure 4, in line 324, change “Control Line” to “Test Line” and “no reactivity” to “invalid test.” In line 325, replace “Test Line” with “Control Line.”

R: Thanks for the suggestion. I have made the suggested corrections in the lines. The 228 and 229 lines, like the 324 and 325 lines.

2. In the statistical analysis section, please remove “positive and negative predictive values” from line 234.

R: Thanks for the suggestion. Remove the positive and negative predictive values. Now the new sentence is, “Sensitivity, specificity, and confidence interval parameters were estimated with the software MedCalc (version 12.3) (MedCalc Software, Ostend, Belgium)”.

3. Also in the statistical analysis, in lines 239 and 240, please add the phrase “optical density.” Specifically, include the following: “The cut-off was determined by adding twice the standard deviation of the negative sera optical density to the mean optical density of the negative samples, corresponding to a 95% confidence interval.”

R: Thanks for the suggestion. As suggested, the following text has been included in lines 239 and 240. The cut-off was determined by adding twice the standard deviation of the negative sera optical density to the mean optical density of the negative samples, corresponding to a 95% confidence interval.

REVIEWER: 2

Thank you for your comments and help in improving the understanding of the text.

---

## [Reviewer Report · REVIEWERS COMMENTS]

## REVIEWER #1

Dear Authors,

Thank you for making the corrections.

Kind regards,

## REVIEWER #2

MIOC-2025-0126.R2

Lines 1-2: Evaluation of a Q5 Chimeric Protein in Rapid Diagnostic Test for Human and Canine Visceral Leishmaniasis

Comments in Methodology and results;

Line 204: Replace ELISA assays with ELISA assays using recombinant proteins (rELISA)

Line 212: Replace RDT optimization with RDT based on the recombinant Q5

Line 230: Replace Multiple tests were then set up.... with Tests were then set up....

Line 247: Replace Reassessment of the ELISA-Q5 for the diagnosis of human VL with ELISA-Q5 for the diagnosis of human VL

Line 281: Replace Reassessment of the ELISA-Q5 for the canine VL diagnosis with ELISA-Q5 for the canine VL diagnosis

Line 309: Replace A new RDT based on the recombinant Q5 with RDT based on the recombinant Q5

Line 410: Replace A direct comparison of the Q5 performance using the ELISA and RDT tests with Comparative analysis of the diagnostic performance of Q5 using ELISA and RDT

Remove lines 248–250, as this information has already been presented in the Materials and Methods section (lines 160–164). “Our first analysis of the recombinant Q5 protein for the diagnosis of human VL using ELISA tested a total of 50 positive sera, all parasitologically confirmed from the Brazilian State of Piauí (PI), plus 50 VL-negative sera (Santos et al. 2020).”

Move Figure 1 to the Materials and Methods section, at the end of the sample description (line 193). Lines 253-254(the origins of all the sera used here and in the following topics are represented in Fig. 1A)

The cutoff value was not presented in the results for canine VL and Human (Fig.2A).

Remove lines 282-284, as this information has already been presented in the Materials and Methods section “The previous analysis of the ELISA-Q5 for diagnosing canine VL evaluated 39 VL-positive sera from parasitologically confirmed dogs from Bahia, all of which produced positive results (Santos et al. 2020).”

Move lines 309–329 to the Materials and Methods section, after the RDT optimization subsection.

Lines 334-335: results shown in Fig. 5A. The RDT-Q5 was first evaluated with a limited panel of 40 VL-positive human sera from Piaui. Replace 40 with 41, as indicated in the Materials and Methods section.

Line 339–340 and Line 387: delete “different” and replace with “from three Brazilian States (Piauí, Mato Grosso do Sul, and Pernambuco)”. .....sera from four Brazilian States, including those sera already assessed with the...

Line 341: replace “healthy” with “negative”, so it reads “A total of 64 sera from negative controls”.

Lines 360 e 382: replace “healthy” with “negative” controls”.

Lines 437–438: Replace with “Table 1. Agreement between the performance of ELISA-Q5 and RDT-Q5 using VL-positive and VL-negative human (HVL) and canine (CVL) sera.”

Line 483: viable alternative for the VL diagnosis, with potential use for both forms of the disease. Replace both forms of the disease with for the diagnosis of human or canine VL.

Line 488: sera qualifies it as a cost-effective alternative for the diagnosis of human disease. Replace sera qualifies it as a cost-effective alternative for the diagnosis of human infection.

Comments in discussion: The occurrence of cross-reactivity with serum samples from dogs diagnosed with cutaneous leishmaniasis has been reported recently and represents a relevant challenge for the interpretation of serological tests. This finding may be associated with antigenic overlap among *Leishmania* species, particularly in light of isolated reports of American cutaneous leishmaniasis (ACL) caused by *Leishmania (L.) infantum*. These cases have been described in patients with or without HIV coinfection in the Central-West and Southwestern regions of Brazil (DOI: 10.1016/j.abd.2020.02.003). This epidemiological scenario reinforces the need for caution in interpreting serological results, as the atypical circulation of *L. infantum* in cutaneous presentations may contribute to cross-reactivity, affecting test specificity and highlighting the importance of combining diagnostic methods for accurate differentiation of leishmaniasis clinical forms.

## AUTHORS’ RESPONSE TO THE REVIEWERS

Point-by-Point Response to Comments_V3

Thank you very much for taking the time to review this manuscript. Please find detailed responses below, along with the corresponding revisions and corrections highlighted in track changes in the resubmitted files.

REVIEWER: 2

General Comments

1. Line 204: Replace ELISA assays with ELISA assays using recombinant proteins (rELISA).

R: Thanks for the suggestion. I have made the suggested corrections in line 204: ( ELISA assays using recombinant proteins (rELISA) .

2. Line 212: Replace RDT optimization with RDT based on the recombinant Q5.

R: Thanks for the suggestion. I have made the suggested corrections in line 212: RDT based on the recombinant. I didn’t specifically put Q5, because in this topic it was also done with other recombinants (Lci2 and Lci13).

3. Line 230: Replace Multiple tests were then set up.... with Tests were then set up....

R: Thanks for the suggestion. I have made the suggested corrections in line 230.

4. Line 247: Replace Reassessment of the ELISA-Q5 for the diagnosis of human VL with ELISA-Q5 for the diagnosis of human VL.

R: Thanks for the suggestion. I have made the suggested corrections in new line 243: ELISA-Q5 for the diagnosis of human VL.

5. Line 281: Replace Reassessment of the ELISA-Q5 for the canine VL diagnosis with ELISA-Q5 for the canine VL diagnosis.

R: Thanks for the suggestion. I have made the suggested corrections in new line 278: ELISA-Q5 for the canine VL diagnosis.

6. Line 309: Replace A new RDT based on the recombinant Q5 with RDT based on the recombinant Q5.

R: Thanks for the suggestion. I have made the suggested corrections in new line 307: RDT based on the recombinant Q5.

7. Line 410: Replace A direct comparison of the Q5 performance using the ELISA and RDT tests with Comparative analysis of the diagnostic performance of Q5 using ELISA and RDT.

R: Thanks for the suggestion. I have made the suggested corrections in new line 408: Comparative analysis of the diagnostic performance of Q5 using ELISA and RDT.

8. Remove lines 248–250, as this information has already been presented in the Materials and Methods section (lines 160–164).

“Our first analysis of the recombinant Q5 protein for the diagnosis of human VL using ELISA tested a total of 50 positive sera, all parasitologically confirmed from the Brazilian State of Piauí (PI), plus 50 VL-negative sera (Santos et al. 2020).

R: Thanks for the suggestion. Remove the “Our first analysis of the recombinant Q5 protein for the diagnosis of human VL using ELISA tested a total of 50 positive sera, all parasitologically confirmed from the Brazilian State of Piauí (PI), plus 50 VL-negative sera (Santos et al. 2020)”. Now the new sentence is, “Thirty-one sera from Pernambuco were first tested with six producing false negative results and defining a sensitivity of 81%. Another 30 sera from Mato Grosso do Sul were also tested, with three false negative results and 90% sensitivity. To determine specificity, 64 VL-negative sera were also tested, with only one producing a false positive result (Fig. 2A and 2B).” Lines 252 – 257.

9. Move Figure 1 to the Materials and Methods section, at the end of the sample description (line 193).

Lines 253-254 (the origins of all the sera used here and in the following topics are represented in Fig. 1A).

R: Thanks for the suggestion. As suggested, the following text has been included in new lines 192 and 193 (the origins of all the sera used here and in the following topics are represented in Fig. 1A). As suggested, move Figure 1 to the Materials and Methods section (Line 195).

10. The cutoff value was not presented in the results for canine VL and Human (Fig.2A).

R: Thanks for the suggestion. As suggested, the following text has been included in new lines 257 and 258 cutoff value Human (Another 30 sera from Mato Grosso do Sul were also tested, with three false negative results and 90% sensitivity (cutoff 0.089). 300 cutoff value Human, to determine specificity, we also tested 47 sera with a previously defined negative diagnosis, of which only one was a false positive (cutoff 0.286).

11. Remove lines 282-284, as this information has already been presented in the Materials and Methods section “The previous analysis of the ELISA-Q5 for diagnosing canine VL evaluated 39 VL-positive sera from parasitologically confirmed dogs from Bahia, all of which produced positive results (Santos et al. 2020).”

R: Thanks for the suggestion. As suggested, the following text has been removing the line “The previous analysis of the ELISA-Q5 for diagnosing canine VL evaluated 39 VL-positive sera from parasitologically confirmed dogs from Bahia, all of which produced positive results (Santos et al. 2020).” Now the new sentence is, “In this study, we reassessed the ELISA-Q5 using a significantly larger sample of VL-positive canine sera (summarized in Fig. 1B).”

12. Move lines 309–329 to the Materials and Methods section, after the RDT optimization subsection.

R: Thanks for the suggestion. We thank the reviewer for the suggestion. However, we chose to keep lines 309–329 in the Results section because the manuscript is primarily focused on the description of a new methodology (RDT-Q5). In this context, these data represent experimental outcomes obtained during the RDT optimization process rather than purely procedural steps. Therefore, we believe their placement in the Results section better reflects their role in demonstrating the performance and development of the assay.

13. Lines 334-335: results shown in Fig. 5A. The RDT-Q5 was first evaluated with a limited panel of 40 VL-positive human sera from Piaui. Replace 40 with 41, as indicated in the Materials and Methods section.

R: Thanks for the suggestion. I have made the suggested corrections in line 339.

14. Line 339–340 and Line 387: delete “different” and replace with “from three Brazilian States (Piauí, Mato Grosso do Sul, and Pernambuco)”. ....sera from four Brazilian States, including those sera already assessed with the....

R: Thanks for the suggestion. I have made the suggested corrections in lines 345 and 392.

15. Line 341: replace “healthy” with “negative”, so it reads “A total of 64 sera from negative controls”.

Lines 360 e 382: replace “healthy” with “negative” controls”.

R: Thanks for the suggestion. I have made the suggested corrections in lines 346, 365 and 387.

16. Lines 437–438: Replace with “Table 1. Agreement between the performance of ELISA-Q5 and RDT-Q5 using VL-positive and VL-negative human (HVL) and canine (CVL) sera.”

R: Thanks for the suggestion. I have made the suggested corrections in new line 443: Table 1. Agreement between the performance of ELISA-Q5 and RDT-Q5 using VL-positive and VL-negative human (HVL) and canine (CVL) sera.

17. Line 483: viable alternative for the VL diagnosis, with potential use for both forms of the disease. Replace both forms of the disease with for the diagnosis of human or canine VL.

R: Thanks for the suggestion. I have made the suggested corrections in new line 490: both forms of the disease with for the diagnosis of human or canine VL.

18. Line 488: sera qualifies it as a cost-effective alternative for the diagnosis of human disease. Replace sera qualifies it as a cost-effective alternative for the diagnosis of human infection.

R: Thanks for the suggestion. I have made the suggested corrections in new line 496: sera qualify it as a cost-effective alternative for the diagnosis of human infection.